# Global Convergence of Langevin Dynamics Based Algorithms for Nonconvex Optimization

**Pan Xu**[*]
Department of Computer Science
UCLA
Los Angeles, CA 90095
panxu@cs.ucla.edu

**Jinghui Chen**[*]
Department of Computer Science
University of Virginia
Charlottesville, VA 22903
jc4zg@virginia.edu

**Difan Zou**
Department of Computer Science
UCLA
Los Angeles, CA 90095
knowzou@cs.ucla.edu

**Quanquan Gu**
Department of Computer Science
UCLA
Los Angeles, CA 90095
qgu@cs.ucla.edu

## Abstract

We present a unified framework to analyze the global convergence of Langevin dynamics based algorithms for nonconvex finite-sum optimization with $n$ component functions. At the core of our analysis is a direct analysis of the ergodicity of the numerical approximations to Langevin dynamics, which leads to faster convergence rates. Specifically, we show that gradient Langevin dynamics (GLD) and stochastic gradient Langevin dynamics (SGLD) converge to the *almost minimizer*[2] within $\widetilde{O}\big(nd/(\lambda\epsilon)\big)$ and $\widetilde{O}\big(d^7/(\lambda^5\epsilon^5)\big)$ stochastic gradient evaluations respectively[3], where $d$ is the problem dimension, and $\lambda$ is the spectral gap of the Markov chain generated by GLD. Both results improve upon the best known gradient complexity[4] results [45]. Furthermore, for the first time we prove the global convergence guarantee for variance reduced stochastic gradient Langevin dynamics (SVRG-LD) to the almost minimizer within $\widetilde{O}\big(\sqrt{n}d^5/(\lambda^4\epsilon^{5/2})\big)$ stochastic gradient evaluations, which outperforms the gradient complexities of GLD and SGLD in a wide regime. Our theoretical analyses shed some light on using Langevin dynamics based algorithms for nonconvex optimization with provable guarantees.

## 1 Introduction

We consider the following nonconvex finite-sum optimization problem

$$\min_{\mathbf{x}} F_n(\mathbf{x}) := 1/n \sum_{i=1}^n f_i(\mathbf{x}), \tag{1.1}$$

where $f_i(\mathbf{x})$'s are called component functions, and both $F_n(\mathbf{x})$ and $f_i(\cdot)$'s can be nonconvex. Various first-order optimization algorithms such as gradient descent [42], stochastic gradient descent [27] and more recently variance-reduced stochastic gradient descent [46, 3] have been proposed and analyzed for solving (1.1). However, all these algorithms are only guaranteed to converge to a stationary point, which can be a local minimum, a local maximum, or even a saddle point. This raises an important

---

[*]Equal contribution.

[2]Following [45], an almost minimizer is defined to be a point which is within the ball of the global minimizer with radius $O(d\log(\beta+1)/\beta)$, where $d$ is the problem dimension and $\beta$ is the inverse temperature parameter.

[3]$\widetilde{O}(\cdot)$ notation hides polynomials of logarithmic terms and constants.

[4]Gradient complexity is defined as the total number of stochastic gradient evaluations of an algorithm, which is the number of stochastic gradients calculated per iteration times the total number of iterations.

question in nonconvex optimization and machine learning: is there an efficient algorithm that is guaranteed to converge to the global minimum of (1.1)?

Recent studies [17, 16] showed that sampling from a distribution which concentrates around the global minimum of $F_n(\mathbf{x})$ is a similar task as minimizing $F_n$ via certain optimization algorithms. This justifies the use of Langevin dynamics based algorithms for optimization. In detail, the first order Langevin dynamics is defined by the following stochastic differential equation (SDE)

$$d\boldsymbol{X}(t) = -\nabla F_n(\boldsymbol{X}(t))dt + \sqrt{2\beta^{-1}}d\boldsymbol{B}(t), \tag{1.2}$$

where $\beta > 0$ is the inverse temperature parameter that is treated as a constant throughout the analysis of this paper, and $\{\boldsymbol{B}(t)\}_{t\geq 0}$ is the standard Brownian motion in $\mathbb{R}^d$. Under certain assumptions on the drift coefficient $\nabla F_n$, it was showed that the distribution of diffusion $\boldsymbol{X}(t)$ in (1.2) converges to its stationary distribution [14], a.k.a., the Gibbs measure $\pi(d\mathbf{x}) \propto \exp(-\beta F_n(\mathbf{x}))$, which concentrates on the global minimum of $F_n$ [29, 26, 47]. Note that the above convergence result holds even when $F_n(\mathbf{x})$ is nonconvex. This motivates the use of Langevin dynamics based algorithms for nonconvex optimization [45, 53, 50, 49]. However, unlike first order optimization algorithms [42, 27, 46, 3], which have been extensively studied, the non-asymptotic theoretical guarantee of applying Langevin dynamics based algorithms for nonconvex optimization, is still under studied. In a seminal work, Raginsky et al. [45] provided a non-asymptotic analysis of stochastic gradient Langevin dynamics (SGLD) [52] for nonconvex optimization, which is a stochastic gradient based discretization of (1.2). They proved that SGLD converges to an almost minimizer up to $d^2/(\sigma^{1/4}\lambda^*)\log(1/\epsilon)$ within $\widetilde{O}(d/(\lambda^*\epsilon^4))$ iterations, where $\sigma^2$ is the variance of stochastic gradient and $\lambda^*$ is called the *uniform spectral gap* of Langevin diffusion (1.2), and it is in the order of $e^{-\widetilde{O}(d)}$. In a concurrent work, Zhang et al. [53] analyzed the hitting time of SGLD and proved its convergence to an approximate local minimum. More recently, Tzen et al. [50] studied the local optimality and generalization performance of Langevin algorithm for nonconvex functions through the lens of metastability and Simsekli et al. [49] developed an asynchronous-parallel stochastic L-BFGS algorithm for non-convex optimization based on variants of SGLD. Erdogdu et al. [23] further developed non-asymptotic analysis of global optimization based on a broader class of diffusions.

In this paper, we establish the global convergence for a family of Langevin dynamics based algorithms, including Gradient Langevin Dynamics (GLD) [17, 20, 16], Stochastic Gradient Langevin Dynamics (SGLD) [52] and Stochastic Variance Reduced Gradient Langevin Dynamics (SVRG-LD) [19] for solving the finite sum nonconvex optimization problem in (1.1). Our analysis is built upon the direct analysis of the discrete-time Markov chain rather than the continuous-time Langevin diffusion, and therefore avoid the discretization error.

## 1.1 Our Contributions

The major contributions of our work are summarized as follows:

- We provide a unified analysis for a family of Langevin dynamics based algorithms by a new decomposition scheme of the optimization error, under which we directly analyze the ergodicity of numerical approximations for Langevin dynamics (see Figure 1).

- Under our unified framework, we establish the global convergence of GLD for solving (1.1). In detail, GLD requires $\widetilde{O}(d/(\lambda\epsilon))$ iterations to converge to the almost minimizer of (1.1) up to precision $\epsilon$, where $\lambda$ is the spectral gap of the discrete-time Markov chain generated by GLD and is in the order of $e^{-\widetilde{O}(d)}$. This improves the $\widetilde{O}(d/(\lambda^*\epsilon^4))$ iteration complexity of GLD implied by [45], where $\lambda^* = e^{-\widetilde{O}(d)}$ is the spectral gap of Langevin diffusion (1.2).

- We establish a faster convergence of SGLD to the almost minimizer of (1.1). In detail, it converges to the almost minimizer up to $\epsilon$ precision within $\widetilde{O}(d^7/(\lambda^5\epsilon^5))$ stochastic gradient evaluations. This also improves the $\widetilde{O}(d^9/(\lambda^{*5}\epsilon^8))$ gradient complexity proved in [45].

- We also analyze the SVRG-LD algorithm and investigate its global convergence property. We show that SVRG-LD is guaranteed to converge to the almost minimizer of (1.1) within $\widetilde{O}(\sqrt{n}d^5/(\lambda^4\epsilon^{5/2}))$ stochastic gradient evaluations. It outperforms the gradient complexities of both GLD and SGLD when $1/\epsilon^3 \leq n \leq 1/\epsilon^5$. To the best of our knowledge, this is the first global convergence guarantee of SVRG-LD for nonconvex optimization, while the original paper [19] only analyzed the posterior sampling property of SVRG-LD.

## 1.2 Additional Related Work

Stochastic gradient Langevin dynamics (SGLD) [52] and its extensions [2, 39, 19] have been widely used in Bayesian learning. A large body of work has focused on analyzing the mean square error of Langevin dynamics based algorithms. In particular, Vollmer et al. [51] analyzed the non-asymptotic bias and variance of the SGLD algorithm by using Poisson equations. Chen et al. [12] showed the non-asymptotic bias and variance of MCMC algorithms with high order integrators. Dubey et al. [19] proposed variance-reduced algorithms based on stochastic gradient Langevin dynamics, namely SVRG-LD and SAGA-LD, for Bayesian posterior inference, and proved that their method improves the mean square error upon SGLD. Li et al. [37] further improved the mean square error by applying the variance reduction tricks on Hamiltonian Monte Carlo, which is also called the underdamped Langevin dynamics.

Another line of research [17, 21, 16, 18, 22, 55] focused on characterizing the distance between distributions generated by Langevin dynamics based algorithms and (strongly) log-concave target distributions. In detail, Dalalyan [17] proved that the distribution of the last step in GLD converges to the stationary distribution in $\widetilde{O}(d/\epsilon^2)$ iterations in terms of total variation distance and Wasserstein distance respectively with a warm start and showed the similarities between posterior sampling and optimization. Later Durmus and Moulines [20] improved the results by showing this result holds for any starting point and established similar bounds for the Wasserstein distance. Dalalyan [16] further improved the existing results in terms of the Wasserstein distance and provide further insights on the close relation between approximate sampling and gradient descent. Cheng et al. [13] improved existing 2-Wasserstein results by reducing the discretization error using underdamped Langevin dynamics. To improve the convergence rates in noisy gradient settings, Chatterji et al. [11], Zou et al. [56] presented convergence guarantees in 2-Wasserstein distance for SAGA-LD and SVRG-LD using variance reduction techniques. Zou et al. [55] proposed the variance reduced Hamilton Monte Carlo to accelerate the convergence of Langevin dynamics based sampling algorithms. As to sampling from distribution with compact support, Bubeck et al. [8] analyzed sampling from log-concave distributions via projected Langevin Monte Carlo, and Brosse et al. [7] proposed a proximal Langevin Monte Carlo algorithm. This line of research is orthogonal to our work since their analyses are regarding to the convergence of the distribution of the iterates to the stationary distribution of Langevin diffusion in total variation distance or 2-Wasserstein distance instead of expected function value gap.

On the other hand, many attempts have been made to escape from saddle points in nonconvex optimization, such as cubic regularization [43, 54], trust region Newton method [15], Hessian-vector product based methods [1, 9, 10], noisy gradient descent [24, 31, 32] and normalized gradient [36]. Yet all these algorithms are only guaranteed to converge to an approximate local minimum rather than a global minimum. The global convergence for nonconvex optimization remains understudied.

## 1.3 Notation and Preliminaries

In this section, we present notations used in this paper and some preliminaries for SDE. We use lower case bold symbol $\mathbf{x}$ to denote deterministic vector, and use upper case italicized bold symbol $\boldsymbol{X}$ to denote random vector. For a vector $\mathbf{x} \in \mathbb{R}^d$, we denote by $\|\mathbf{x}\|_2$ its Euclidean norm. We use $a_n = O(b_n)$ to denote that $a_n \leq C b_n$ for some constant $C > 0$ independent of $n$. We also denote $a_n \lesssim b_n$ ($a_n \gtrsim b_n$) if $a_n$ is less than (larger than) $b_n$ up to a constant. We also use $\widetilde{O}(\cdot)$ notation to hide both polynomials of logarithmic terms and constants.

**Kolmogorov Operator and Infinitesimal Generator**
Suppose $\boldsymbol{X}(t)$ is the solution to the diffusion process represented by the stochastic differential equation (1.2). For such a continuous time Markov process, let $P = \{P_t\}_{t>0}$ be the corresponding Markov semi-group [4], and we define the Kolmogorov operator [4] $P_s$ as follows

$$P_s g(\boldsymbol{X}(t)) = \mathbb{E}[g(\boldsymbol{X}(s+t))|\boldsymbol{X}(t)],$$

where $g$ is a smooth test function. We have $P_{s+t} = P_s \circ P_t$ by Markov property. Further we define the infinitesimal generator [4] of the semi-group $\mathcal{L}$ to describe the the movement of the process in an infinitesimal time interval:

$$\mathcal{L}g(\boldsymbol{X}(t)) := \lim_{h \to 0^+} \frac{\mathbb{E}[g(\boldsymbol{X}(t+h))|\boldsymbol{X}(t)] - g(\boldsymbol{X}(t))}{h} = \big(-\nabla F_n(\boldsymbol{X}(t)) \cdot \nabla + \beta^{-1}\nabla^2\big)g(\boldsymbol{X}(t)),$$

where $\beta$ is the inverse temperature parameter.

**Poisson Equation and the Time Average**

Poisson equations are widely used in the study of homogenization and ergodic theory to prove the desired limit of a time-average. Let $\mathcal{L}$ be the infinitesimal generator and let $\psi$ be defined as follows

$$\mathcal{L}\psi = g - \bar{g}, \qquad (1.3)$$

where $g$ is a smooth test function and $\bar{g}$ is the expectation of $g$ over the Gibbs measure, i.e., $\bar{g} := \int g(\mathbf{x})\pi(d\mathbf{x})$. Smooth function $\psi$ is called the solution of Poisson equation (1.3). Importantly, it has been shown [23] that the first and second order derivatives of the solution $\psi$ of Poisson equation for Langevin diffusion can be bounded by polynomial growth functions.

## 2 Review of Langevin Dynamics Based Algorithms

In this section, we briefly review three Langevin dynamics based algorithms proposed recently.

In practice, numerical methods (a.k.a., numerical integrators) are used to approximate the Langevin diffusion in (1.2). For example, by Euler-Maruyama scheme [34], (1.2) can be discretized as follows:

$$\boldsymbol{X}_{k+1} = \boldsymbol{X}_k - \eta\nabla F_n(\boldsymbol{X}_k) + \sqrt{2\eta\beta^{-1}}\cdot\boldsymbol{\epsilon}_k, \qquad (2.1)$$

where $\boldsymbol{\epsilon}_k \in \mathbb{R}^d$ is standard Gaussian noise and $\eta > 0$ is the step size. The update in (2.1) resembles gradient descent update except for an additional injected Gaussian noise. The magnitude of the Gaussian noise is controlled by the inverse temperature parameter $\beta$. In our paper, we refer this update as Gradient Langevin Dynamics (GLD) [17, 20, 16]. The details of GLD are shown in Algorithm 1.

In the case that $n$ is large, the above Euler approximation can be infeasible due to the high computational cost of the full gradient $\nabla F_n(\boldsymbol{X}_k)$ at each iteration. A natural idea is to use stochastic gradient to approximate the full gradient, which gives rise to Stochastic Gradient Langevin Dynamics (SGLD) [52] and its variants [2, 39, 12]. However, the high variance brought by the stochastic gradient can make the convergence of SGLD slow. To reduce the variance of the stochastic gradient and accelerate the convergence of SGLD, we use a mini-batch of stochastic gradients in the following update form:

$$\boldsymbol{Y}_{k+1} = \boldsymbol{Y}_k - \eta/B\sum_{i\in I_k}\nabla f_i(\boldsymbol{Y}_k) + \sqrt{2\eta\beta^{-1}}\cdot\boldsymbol{\epsilon}_k, \qquad (2.2)$$

where $1/B\sum_{i\in I_k}\nabla f_i(\boldsymbol{Y}_k)$ is the stochastic gradient, which is an unbiased estimator for $\nabla F_n(\boldsymbol{Y}_k)$ and $I_k$ is a subset of $\{1,\dots,n\}$ with $|I_k| = B$. Algorithm 2 displays the details of SGLD.

Motivated by recent advances in stochastic optimization, in particular, the variance reduction based techniques [33, 46, 3], Dubey et al. [19] proposed the Stochastic Variance Reduced Gradient Langevin Dynamics (SVRG-LD) for posterior sampling. The key idea is to use semi-stochastic gradient to reduce the variance of the stochastic gradient. SVRG-LD takes the following update form:

$$\boldsymbol{Z}_{k+1} = \boldsymbol{Z}_k - \eta\widetilde{\nabla}_k + \sqrt{2\eta\beta^{-1}}\cdot\boldsymbol{\epsilon}_k, \qquad (2.3)$$

where $\widetilde{\nabla}_k = 1/B\sum_{i_k\in I_k}\big(\nabla f_{i_k}(\boldsymbol{Z}_k) - \nabla f_{i_k}(\widetilde{\boldsymbol{Z}}^{(s)}) + \nabla F_n(\widetilde{\boldsymbol{Z}}^{(s)})\big)$ is the semi-stochastic gradient, $\widetilde{\boldsymbol{Z}}^{(s)}$ is a snapshot of $\boldsymbol{Z}_k$ at every $L$ iteration such that $k = sL + \ell$ for some $\ell = 0, 1, \dots, L-1$, and $I_k$ is a subset of $\{1,\dots,n\}$ with $|I_k| = B$. SVRG-LD is summarized in Algorithm 3.

Note that although all the three algorithms are originally proposed for posterior sampling or more generally, Bayesian learning, they can be applied for nonconvex optimization, as demonstrated in many previous studies [2, 45, 53].

---

**Algorithm 1** Gradient Langevin Dynamics (GLD)

    **input:** step size $\eta > 0$; inverse temperature parameter $\beta > 0$; $\boldsymbol{X}_0 = \boldsymbol{0}$
    **for** $k = 0, 1, \dots, K-1$ **do**
        randomly draw $\boldsymbol{\epsilon}_k \sim N(\boldsymbol{0}, \mathbf{I}_{d\times d})$
        $\boldsymbol{X}_{k+1} = \boldsymbol{X}_k - \eta\nabla F_n(\boldsymbol{X}_k) + \sqrt{2\eta/\beta}\boldsymbol{\epsilon}_k$
    **end for**

---

## 3 Main Theory

Before we present our main results, we first lay out the following assumptions on the loss function.

**Assumption 3.1** (Smoothness). The function $f_i(\mathbf{x})$ is $M$-smooth for $M > 0$, $i = 1, \dots, n$, i.e.,

$$\|\nabla f_i(\mathbf{x}) - \nabla f_i(\mathbf{y})\|_2 \le M\|\mathbf{x} - \mathbf{y}\|_2, \quad \text{for any } \mathbf{x}, \mathbf{y} \in \mathbb{R}^d.$$

---

**Algorithm 2** Stochastic Gradient Langevin Dynamics (SGLD)

---
**input:** step size $\eta > 0$; batch size $B$; inverse temperature parameter $\beta > 0$; $\boldsymbol{Y}_0 = \boldsymbol{0}$
**for** $k = 0, 1, \ldots, K - 1$ **do**
    randomly pick a subset $I_k$ from $\{1, \ldots, n\}$ of size $|I_k| = B$; randomly draw $\boldsymbol{\epsilon}_k \sim N(\boldsymbol{0}, \mathbf{I}_{d \times d})$
    $\boldsymbol{Y}_{k+1} = \boldsymbol{Y}_k - \eta/B \sum_{i \in I_k} \nabla f_i(\boldsymbol{Y}_k) + \sqrt{2\eta/\beta}\boldsymbol{\epsilon}_k$
**end for**

---

---

**Algorithm 3** Stochastic Variance Reduced Gradient Langevin Dynamics (SVRG-LD)

---
**input:** step size $\eta > 0$; batch size $B$; epoch length $L$; inverse temperature parameter $\beta > 0$
**initialization:** $\boldsymbol{Z}_0 = \boldsymbol{0}$, $\widetilde{\boldsymbol{Z}}^{(0)} = \boldsymbol{Z}_0$
**for** $s = 0, 1, \ldots, (K/L) - 1$ **do**
    $\widehat{\boldsymbol{W}} = \nabla F_n(\widetilde{\boldsymbol{Z}}^{(s)})$
    **for** $\ell = 0, \ldots, L - 1$ **do**
        $k = sL + \ell$
        randomly pick a subset $I_k$ from $\{1, \ldots, n\}$ of size $|I_k| = B$; draw $\boldsymbol{\epsilon}_k \sim N(\boldsymbol{0}, \mathbf{I}_{d \times d})$
        $\widetilde{\nabla}_k = 1/B \sum_{i_k \in I_k} \left( \nabla f_{i_k}(\boldsymbol{Z}_k) - \nabla f_{i_k}(\widetilde{\boldsymbol{Z}}^{(s)}) + \widehat{\boldsymbol{W}} \right)$
        $\boldsymbol{Z}_{k+1} = \boldsymbol{Z}_k - \eta\widetilde{\nabla}_k + \sqrt{2\eta/\beta}\boldsymbol{\epsilon}_k$
    **end for**
    $\widetilde{\boldsymbol{Z}}^{(s)} = \boldsymbol{Z}_{(s+1)L}$
**end for**

---

Assumption 3.1 immediately implies that $F_n(\mathbf{x}) = 1/n \sum_{i=1}^{n} f_i(\mathbf{x})$ is also $M$-smooth.

**Assumption 3.2** (Dissipative). There exist constants $m, b > 0$, such that we have

$$\langle \nabla F_n(\mathbf{x}), \mathbf{x} \rangle \geq m\|\mathbf{x}\|_2^2 - b, \quad \text{for all } \mathbf{x} \in \mathbb{R}^d.$$

Assumption 3.2 is a typical assumption for the convergence analysis of an SDE and diffusion approximation [40, 45, 53], which can be satisfied by enforcing a weight decay regularization [45]. It says that starting from a position that is sufficiently far away from the origin, the Markov process defined by (1.2) moves towards the origin on average. It can also be noted that all critical points are within the ball of radius $O(\sqrt{b/m})$ centered at the origin under this assumption.

Let $\mathbf{x}^* = \operatorname{argmin}_{\mathbf{x} \in \mathbb{R}^d} F_n(\mathbf{x})$ be the global minimizer of $F_n$. Our ultimate goal is to prove the convergence of the optimization error in expectation, i.e., $\mathbb{E}[F_n(\boldsymbol{X}_k)] - F_n(\mathbf{x}^*)$. In the sequel, we decompose the optimization error into two parts: (1) $\mathbb{E}[F_n(\boldsymbol{X}_k)] - \mathbb{E}[F_n(\boldsymbol{X}^\pi)]$, which characterizes the gap between the expected function value at the $k$-th iterate $\boldsymbol{X}_k$ and the expected function value at $\boldsymbol{X}^\pi$, where $\boldsymbol{X}^\pi$ follows the stationary distribution $\pi(d\mathbf{x})$ of Markov process $\{\boldsymbol{X}(t)\}_{t \geq 0}$, and (2) $\mathbb{E}[F_n(\boldsymbol{X}^\pi)] - F_n(\mathbf{x}^*)$. Note that the error in part (1) is algorithm dependent, while that in part (2) only depends on the diffusion itself and hence is identical for all Langevin dynamics based algorithms.

Now we are ready to present our main results regarding to the optimization error of each algorithm reviewed in Section 2. We first show the optimization error bound of GLD (Algorithm 1).

**Theorem 3.3** (GLD). Under Assumptions 3.1 and 3.2, consider $\boldsymbol{X}_K$ generated by Algorithm 1 with initial point $\boldsymbol{X}_0 = \boldsymbol{0}$. The optimization error is bounded by

$$\mathbb{E}[F_n(\boldsymbol{X}_K)] - F_n(\mathbf{x}^*) \leq \Theta e^{-\lambda K \eta} + \frac{C_\psi \eta}{\beta} + \underbrace{\frac{d}{2\beta} \log\left( \frac{eM(b\beta/d + 1)}{m} \right)}_{\mathcal{R}_M}, \qquad (3.1)$$

where problem-dependent parameters $\Theta$ and $\lambda$ are defined as

$$\Theta = \frac{C_0 M(b\beta + m\beta + d)(m + e^{m\eta}M(b\beta + m\beta + d))}{m^2 \rho^{d/2}}, \quad \lambda = \frac{2m\rho^d}{\log(2M(b\beta + m\beta + d)/m)},$$

and $\rho \in (0, 1)$, $C_0, C_\psi > 0$ are constants.

In the optimization error of GLD (3.1), we denote the upper bound of the error term $\mathbb{E}[F_n(\boldsymbol{X}^\pi)] - F_n(\mathbf{x}^*)$ by $\mathcal{R}_M$, which characterizes the distance between the expected function value at $\boldsymbol{X}^\pi$ and the global minimum of $F_n$. The stationary distribution of Langevin diffusion $\pi \propto e^{-\beta F_n(\mathbf{x})}$ is a Gibbs

distribution, which concentrates around the minimizer $\mathbf{x}^*$ of $F_n$. Thus a random vector $\boldsymbol{X}^\pi$ following the law of $\pi$ is called an *almost minimizer* of $F_n$ within a neighborhood of $\mathbf{x}^*$ with radius $\mathcal{R}_M$ [45].

It is worth noting that the first term in (3.1) vanishes at a exponential rate due to the ergodicity of Markov chain $\{\boldsymbol{X}_k\}_{k=0,1,\dots}$. Moreover, the exponential convergence rate is controlled by $\lambda$, the spectral gap of the discrete-time Markov chain generated by GLD, which is in the order of $e^{-\widetilde{O}(d)}$.

By setting $\mathbb{E}[F_n(\boldsymbol{X}_K)] - \mathbb{E}[F_n(\boldsymbol{X}^\pi)]$ to be less than a precision $\epsilon$, and solving for $K$, we have the following corollary on the iteration complexity for GLD to converge to the almost minimizer $\boldsymbol{X}^\pi$.

**Corollary 3.4** (GLD). Under the same conditions as in Theorem 3.3, provided that $\eta \lesssim \epsilon$, GLD achieves $\mathbb{E}[F_n(\boldsymbol{X}_K)] - \mathbb{E}[F_n(\boldsymbol{X}^\pi)] \leq \epsilon$ with $K = O\big(d\epsilon^{-1}\lambda^{-1} \cdot \log(1/\epsilon)\big)$.

**Remark 3.5.** In a seminal work by [45], they provided a non-asymptotic analysis of SGLD for nonconvex optimization. By setting the variance of stochastic gradient to 0, their result immediately suggests an $O(d/(\epsilon^4\lambda^*) \log^5((1/\epsilon)))$ iteration complexity for GLD to converge to the almost minimizer up to precision $\epsilon$. Here the quantity $\lambda^*$ is the so-called **uniform spectral gap** for continuous-time Markov process $\{\boldsymbol{X}_t\}_{t\geq 0}$ generated by Langevin dynamics. They further proved that $\lambda^* = e^{-\widetilde{O}(d)}$, which is in the same order of our spectral gap $\lambda$ for the discrete-time Markov chain $\{\boldsymbol{X}_k\}_{k=0,1\dots}$ generated by GLD. Both of them match the lower bound for metastable exit times of SDE for nonconvex functions that have multiple local minima and saddle points [6]. Although for some specific function $F_n$, the spectral gap may be reduced to polynomial in $d$ [25], in general, the spectral gap for continuous-time Markov processes is in the same order as the spectral gap for discrete-time Markov chains. Thus, the iteration complexity of GLD suggested by Corollary 3.4 is better than that suggested by [45] by a factor of $O(1/\epsilon^3)$.

We now present the following theorem, which states the optimization error of SGLD (Algorithm 2).

**Theorem 3.6** (SGLD). Under Assumptions 3.1 and 3.2, consider $\boldsymbol{Y}_K$ generated by Algorithm 2 with initial point $\boldsymbol{Y}_0 = \mathbf{0}$, the optimization error is bounded by

$$\mathbb{E}[F_n(\boldsymbol{Y}_K)] - F_n(\mathbf{x}^*) \leq C_1\Gamma K\eta\left[\frac{\beta(n-B)(M\sqrt{\Gamma}+G)^2}{B(n-1)}\right]^{1/2} + \Theta e^{-\lambda K\eta} + \frac{C_\psi\eta}{\beta} + \mathcal{R}_M,$$

(3.2)

where $C_1$ is an absolute constant, $C_\psi, \lambda, \Theta$ and $\mathcal{R}_M$ are the same as in Theorem 3.3, $B$ is the mini-batch size, $G = \max_{i=1,\dots,n}\{\|\nabla f_i(\mathbf{x}^*)\|_2\} + bM/m$ and $\Gamma = 2(1 + 1/m)(b + 2G^2 + d/\beta)$.

Similar to Corollary 3.4, by setting $\mathbb{E}[F_n(\boldsymbol{Y}_k)] - \mathbb{E}[F_n(\boldsymbol{X}^\pi)] \leq \epsilon$, we obtain the following corollary.

**Corollary 3.7** (SGLD). Under the same conditions as in Theorem 3.6, if $\eta \lesssim \epsilon$, SGLD achieves

$$\mathbb{E}[F_n(\boldsymbol{Y}_K)] - \mathbb{E}[F_n(\boldsymbol{X}^\pi)] = O\big(d^{3/2}B^{-1/4}\lambda^{-1} \cdot \log(1/\epsilon) + \epsilon\big),$$

(3.3)

with $K = O\big(d\epsilon^{-1}\lambda^{-1} \cdot \log(1/\epsilon)\big)$, where $B$ is the mini-batch size of Algorithm 2.

**Remark 3.8.** Corollary 3.7 suggests that if the mini-batch size $B$ is chosen to be large enough to offset the divergent term $\log(1/\epsilon)$, SGLD is able to converge to the almost minimizer in terms of expected function value gap. This is also suggested by the result in [45]. More specifically, the result in [45] implies that SGLD achieves

$$\mathbb{E}[F_n(\boldsymbol{Y}_K)] - \mathbb{E}[F_n(\boldsymbol{X}^\pi)] = O\big(d^2\sigma^{-1/4}\lambda^{*-1} \cdot \log(1/\epsilon) + \epsilon\big)$$

with $K = O(d/(\lambda^*\epsilon^4) \cdot \log^5(1/\epsilon))$, where $\sigma^2$ is the upper bound of stochastic variance in SGLD, which can be reduced with larger batch size $B$. Recall that the spectral gap $\lambda^*$ in their work scales as $O(e^{-\widetilde{O}(d)})$, which is in the same order as $\lambda$ in Corollary 3.7. In comparison, our result in Corollary 3.7 indicates that SGLD can actually achieve the same order of error for $\mathbb{E}[F_n(\boldsymbol{Y}_K)] - \mathbb{E}[F_n(\boldsymbol{X}^\pi)]$ with substantially fewer number of iterations, i.e., $O(d/(\lambda\epsilon)\log(1/\epsilon))$.

**Remark 3.9.** To ensure SGLD converges in Corollary 3.7, one may set a sufficiently large batch size $B$ to offset the divergent term. For example, if we choose $B \gtrsim d^6\lambda^{-4}\epsilon^{-4}\log^4(1/\epsilon)$, SGLD achieves $\mathbb{E}[F_n(\boldsymbol{Y}_K)] - \mathbb{E}[F_n(\boldsymbol{X}^\pi)] \leq \epsilon$ within $K = O(d/(\lambda\epsilon)\log(1/\epsilon))$ stochastic gradient evaluations.

In what follows, we proceed to present our result on the optimization error bound of SVRG-LD.

**Theorem 3.10** (SVRG-LD). Under Assumptions 3.1 and 3.2, consider $\boldsymbol{Z}_K$ generated by Algorithm 3 with initial point $\boldsymbol{Z}_0 = \boldsymbol{0}$. The optimization error is bounded by

$$\mathbb{E}[F_n(\boldsymbol{Z}_K)] - F_n(\mathbf{x}^*)$$
$$\leq C_1 \Gamma K^{3/4} \eta \left[ \frac{L\beta M^2(n-B)}{B(n-1)} \left( 9\eta L(M^2\Gamma + G^2) + \frac{d}{\beta} \right) \right]^{1/4} + \Theta e^{-\lambda K \eta} + \frac{C_\psi \eta}{\beta} + \mathcal{R}_M, \quad (3.4)$$

where constants $C_1, C_\psi, \lambda, \Theta, \Gamma, G$ and $\mathcal{R}_M$ are the same as in Theorem 3.6, $B$ is the mini-batch size and $L$ is the length of inner loop of Algorithm 3.

Similar to Corollaries 3.4 and 3.7, we have the following iteration complexity for SVRG-LD.

**Corollary 3.11** (SVRG-LD). Under the same conditions as in Theorem 3.10, if $\eta \lesssim \epsilon$, SVRG-LD achieves $\mathbb{E}[F_n(\boldsymbol{Z}_K)] - \mathbb{E}[F_n(\boldsymbol{X}^\pi)] \leq \epsilon$ within $K = O\big(Ld^5 B^{-1}\lambda^{-4}\epsilon^{-4} \cdot \log^4(1/\epsilon) + 1/\epsilon\big)$ total iterations. In addition, if we choose $B = \sqrt{n}\epsilon^{-3/2}$, $L = \sqrt{n}\epsilon^{3/2}$, the number of stochastic gradient evaluations needed for SVRG-LD to achieve $\epsilon$ precision is $\widetilde{O}\big(\sqrt{n}\epsilon^{-5/2}\big) \cdot e^{\widetilde{O}(d)}$.

**Remark 3.12.** In Theorem 3.10 and Corollary 3.11, we establish the global convergence guarantee for SVRG-LD to an almost minimizer of a nonconvex function $F_n$. To the best of our knowledge, this is the first iteration/gradient complexity guarantee for SVRG-LD in nonconvex finite-sum optimization. Dubey et al. [19] first proposed the SVRG-LD algorithm for posterior sampling, but only proved that the mean square error between averaged sample pass and the stationary distribution converges to $\epsilon$ within $\widetilde{O}(1/\epsilon^{3/2})$ iterations, which has no implication for nonconvex optimization.

In large scale machine learning problems, the evaluation of full gradient can be quite expensive, in which case the iteration complexity is no longer appropriate to reflect the efficiency of different algorithms. To perform a comprehensive

Table 1: Gradient complexities to converge to the almost minimizer.

| | GLD | SGLD[5] | SVRG-LD |
|---|---|---|---|
| [45] | $\widetilde{O}\big(\frac{n}{\epsilon^4}\big) \cdot e^{\widetilde{O}(d)}$ | $\widetilde{O}\big(\frac{1}{\epsilon^8}\big) \cdot e^{\widetilde{O}(d)}$ | N/A |
| This paper | $\widetilde{O}\big(\frac{n}{\epsilon}\big) \cdot e^{\widetilde{O}(d)}$ | $\widetilde{O}\big(\frac{1}{\epsilon^5}\big) \cdot e^{\widetilde{O}(d)}$ | $\widetilde{O}\big(\frac{\sqrt{n}}{\epsilon^{5/2}}\big) \cdot e^{\widetilde{O}(d)}$ |

comparison among the three algorithms, we present their gradient complexities for converging to the almost minimizer $\boldsymbol{X}^\pi$ with $\epsilon$ precision in Table 1. Recall that gradient complexity is defined as the total number of stochastic gradient evaluations needed to achieve $\epsilon$ precision. It can be seen from Table 1 that the gradient complexity for GLD has worse dependence on the number of component functions $n$ and SVRG-LD has worse dependence on the optimization precision $\epsilon$. More specifically, when the number of component functions satisfies $n \leq 1/\epsilon^5$, SVRG-LD achieves better gradient complexity than SGLD. Additionally, if $n \geq 1/\epsilon^3$, SVRG-LD is better than both GLD and SGLD, therefore is more favorable.

## 4 Proof Sketch of the Main Results

In this section, we highlight our high level idea in the analysis of GLD, SGLD and SVRG-LD.

### 4.1 Roadmap of the Proof

Recall the problem in (1.1) and denote the global minimizer as $\mathbf{x}^* = \arg\min_{\mathbf{x}} F_n(\mathbf{x})$. $\{\boldsymbol{X}(t)\}_{t\geq 0}$ and $\{\boldsymbol{X}_k\}_{k=0,\ldots,K}$ are the continuous-time and discrete-time Markov processes generated by Langevin diffusion (1.2) and GLD respectively. We propose to decompose the optimization error as follows:

$$\mathbb{E}[F_n(\boldsymbol{X}_k)] - F_n(\mathbf{x}^*)$$
$$= \underbrace{\mathbb{E}[F_n(\boldsymbol{X}_k)] - \mathbb{E}[F_n(\boldsymbol{X}^\mu)]}_{I_1} + \underbrace{\mathbb{E}[F_n(\boldsymbol{X}^\mu)] - \mathbb{E}[F_n(\boldsymbol{X}^\pi)]}_{I_2} + \underbrace{\mathbb{E}[F_n(\boldsymbol{X}^\pi)] - F_n(\mathbf{x}^*)}_{I_3}, \quad (4.1)$$

where $\boldsymbol{X}^\mu$ follows the stationary distribution $\mu(d\mathbf{x})$ of Markov process $\{\boldsymbol{X}_k\}_{k=0,1,\ldots,K}$, and $\boldsymbol{X}^\pi$ follows the stationary distribution $\pi(d\mathbf{x})$ of Markov process $\{\boldsymbol{X}(t)\}_{t\geq 0}$, a.k.a., the Gibbs distribution. Following existing literature [40, 41, 12], here we assume the existence of stationary distributions, i.e., the ergodicity, of Langevin diffusion (1.2) and its numerical approximation (2.2). Note that the

ergodicity property of an SDE is not trivially guaranteed in general and establishing the existence of the stationary distribution is beyond the scope of our paper. Yet we will discuss the circumstances when geometric ergodicity holds in the Appendix.

We illustrate the decomposition (4.1) in Figure 1. Unlike existing optimization analysis of SGLD such as [45], which measure the approximation error between $\boldsymbol{X}_k$ and $\boldsymbol{X}(t)$ (blue arrows in the chart), we directly analyze the geometric convergence of discretized Markov chain $\boldsymbol{X}_k$ to its stationary distribution (red arrows in the chart). Since the distance between $\boldsymbol{X}_k$ and $\boldsymbol{X}(t)$ is a slow-convergence term in [45], and the distance between $\boldsymbol{X}(t)$ and $\boldsymbol{X}^\pi$ depends on the uniform spectral gap, our new roadmap of proof will bypass both of these two terms, hence leads to a faster convergence rate.



Figure 1: Illustration of the analysis framework in our paper.

**Bounding $I_1$: Geometric Ergodicity of GLD**

To bound the first term in (4.1), we need to analyze the convergence of the Markov chain generated by Algorithm 1 to its stationary distribution, namely, the ergodic property of the numerical approximation of Langevin dynamics. In probability theory, ergodicity describes the long time behavior of Markov processes. For a finite-state Markov Chain, this is also closely related to the mixing time and has been thoroughly studied in the literature of Markov processes [28, 35, 4]. Note that Durmus and Moulines [21] studied the convergence of the Euler-Maruyama discretization (also referred to as the unadjusted Langevin algorithm) towards its stationary distribution in total variation. Nevertheless, they only focus on strongly convex functions which are less challenging than our nonconvex setting.

The following lemma ensures the geometric ergodicity of gradient Langevin dynamics.

**Lemma 4.1.** Under Assumptions 3.1 and 3.2, the gradient Langevin dynamics (GLD) in Algorithm 1 has a unique invariant measure $\mu$ on $\mathbb{R}^d$. It holds that

$$|\mathbb{E}[F_n(\boldsymbol{X}_k)] - \mathbb{E}[F_n(\boldsymbol{X}^\mu)]| \leq C\kappa\rho^{-d/2}(1 + \kappa e^{m\eta}) \exp\left(-\frac{2mk\eta\rho^d}{\log(\kappa)}\right),$$

where $\rho \in (0,1)$, $C > 0$ are absolute constants, and $\kappa = 2M(b\beta + m\beta + d)/b$.

Lemma 4.1 establishes the exponential decay of function gap between $F_n(\boldsymbol{X}_k)$ and $F_n(\boldsymbol{X}^\pi)$ using coupling techniques. Note that the exponential dependence on dimension $d$ is consistent with the result from [45] using entropy methods.

**Bounding $I_2$: Convergence to Stationary Distribution of Langevin Diffusion**

Now we are going to bound the distance between two invariant measures $\mu$ and $\pi$ in terms of their expectations over the objective function $F_n$. Our proof is inspired by [51, 12]. The key insight here is that after establishing the geometric ergodicity of GLD, by the stationarity of $\mu$, we have

$$\int F_n(\mathbf{x})\mu(d\mathbf{x}) = \int \mathbb{E}[F_n(\boldsymbol{X}_k)|\boldsymbol{X}_0 = \mathbf{x}] \cdot \mu(d\mathbf{x}).$$

This property says that after reaching the stationary distribution, any further transition (GLD update) will not change the distribution. Thus we can bound the difference between two invariant measures.

**Lemma 4.2.** Under Assumptions 3.1 and 3.2, the invariant measures $\mu$ and $\pi$ satisfy

$$\left|\mathbb{E}[F_n(\boldsymbol{X}^\mu)] - \mathbb{E}[F_n(\boldsymbol{X}^\pi)]\right| \leq C_\psi \eta/\beta,$$

where $C_\psi > 0$ is a constant that dominates $\mathbb{E}[\|\nabla^p \psi(\boldsymbol{X}_k)\|]$ ($p = 0, 1, 2$) and $\psi$ is the solution of Poisson equation (1.3).

Lemma 4.2 suggests that the bound on the difference between the two invariant measures depends on the numerical approximation step size $\eta$, the inverse temperature parameter $\beta$ and the upper bound $C_\psi$. We emphasize that the dependence on $\beta$ is reasonable since different $\beta$ results in different diffusion, and further leads to different stationary distributions of the SDE and its numerical approximations.

**Bounding $I_3$: Gap between Langevin Diffusion and Global Minimum**

Most existing studies [52, 48, 12] on Langevin dynamics based algorithms focus on the convergence of the averaged sample path to the stationary distribution. The property of Langevin diffusion

asymptotically concentrating on the global minimum of $F_n$ is well understood [14, 26] , which makes the convergence to a global minimum possible, even when the function $F_n$ is nonconvex.

We give an explicit bound between the stationary distribution of Langevin diffusion and the global minimizer of $F_n$, i.e., the last term $\mathbb{E}[F_n(\boldsymbol{X}^{\pi})] - F_n(\mathbf{x}^*)$ in (4.1). For nonconvex objective function, this has been proved in [45] using the concept of differential entropy and smoothness of $F_n$. We formally summarize it as the following lemma:

**Lemma 4.3.** [45] Under Assumptions 3.1 and 3.2, the model error $I_3$ in (4.1) can be bounded by

$$\mathbb{E}[F_n(\boldsymbol{X}^{\pi})] - F_n(\mathbf{x}^*) \leq \frac{d}{2\beta} \log \left( \frac{eM(m\beta/d + 1)}{m} \right),$$

where $\boldsymbol{X}^{\pi}$ is a random vector following the stationary distribution of Langevin diffusion (1.2).

Lemma 4.3 suggests that Gibbs density concentrates on the global minimizer of objective function. Therefore, the random vector $\boldsymbol{X}^{\pi}$ following the Gibbs distribution $\pi$ is also referred to as an *almost minimizer* of the nonconvex function $F_n$ in [45].

### 4.2 Proof of Theorems 3.3, 3.6 and 3.10

Now we integrate the previous lemmas to prove our main theorems in Section 3. First, submitting the results in Lemmas 4.1, 4.2 and 4.3 into (4.1), we immediately obtain the optimization error bound in (3.1) for GLD, which proves Theorem 3.3. Second, consider the optimization error of SGLD (Algorithm 2), we only need to bound the error between $\mathbb{E}[F_n(\boldsymbol{Y}_K)]$ and $\mathbb{E}[F_n(\boldsymbol{X}_K)]$ and then apply the results for GLD, which is given by the following lemma.

**Lemma 4.4.** Under Assumptions 3.1 and 3.2, by choosing mini-batch of size $B$, the output of SGLD in Algorithm 2 ($\boldsymbol{Y}_K$) and the output of GLD in Algorithm 1 ($\boldsymbol{X}_K$) satisfies

$$|\mathbb{E}[F_n(\boldsymbol{Y}_K)] - \mathbb{E}[F_n(\boldsymbol{X}_K)]| \leq C_1 \sqrt{\beta}\Gamma(M\sqrt{\Gamma} + G)K\eta \left[ \frac{n-B}{B(n-1)} \right]^{1/4}, \qquad (4.2)$$

where $C_1$ is an absolute constant and $\Gamma = 2(1 + 1/m)(b + 2G^2 + d/\beta)$.

Combining Lemmas 4.1, 4.2, 4.3 and 4.4 yields the desired result in (3.6) for SGLD, which completes the proof of Theorem 3.6. Third, similar to the proof of SGLD, we require an additional bound between $F_n(\boldsymbol{Z}_K)$ and $F_n(\boldsymbol{X}_K)$ for the proof of SVRG-LD, which is stated by the following lemma.

**Lemma 4.5.** Under Assumptions 3.1 and 3.2, by choosing mini-batch of size $B$, the output of SVRG-LD in Algorithm 3 ($\boldsymbol{Z}_K$) and the output of GLD in Algorithm 1 ($\boldsymbol{X}_K$) satisfies

$$\left| \mathbb{E}[F_n(\boldsymbol{Z}_K)] - \mathbb{E}[F_n(\boldsymbol{X}_K)] \right| \leq C_1 \Gamma K^{3/4}\eta \left[ \frac{LM^2(n-B)(3L\eta\beta(M^2\Gamma + G^2) + d/2)}{B(n-1)} \right]^{1/4},$$

where $\Gamma = 2(1 + 1/m)(b + 2G^2 + d/\beta)$, $C_1$ is an absolute constant and $L$ is the number of inner loops in SVRG-LD.

The optimization error bound in (3.4) for SVRG-LD follows from Lemmas 4.1, 4.2, 4.3 and 4.5.

## 5 Conclusions and Future Work

In this work, we present a new framework for analyzing the convergence of Langevin dynamics based algorithms, and provide non-asymptotic analysis on the convergence for nonconvex finite-sum optimization. By comparing the Langevin dynamics based algorithms and standard first-order optimization algorithms, we may see that the counterparts of GLD and SVRG-LD are gradient descent (GD) and stochastic variance-reduced gradient (SVRG) methods. It has been proved that SVRG outperforms GD universally for nonconvex finite-sum optimization [46, 3]. This poses a natural question that whether SVRG-LD can be universally better than GLD for nonconvex optimization? We will attempt to answer this question in the future.

### Acknowledgement

We would like to thank the anonymous reviewers for their helpful comments. We thank Maxim Raginsky for insightful comments and discussion on the first version of this paper. We also thank Tianhao Wang for discussion on this work. This research was sponsored in part by the National Science Foundation IIS-1652539. The views and conclusions contained in this paper are those of the authors and should not be interpreted as representing any funding agencies.

## Footnotes

[5]For SGLD in [45], the result in the table is obtained by choosing the exact batch size suggested by the authors that could make the stochastic variance small enough to cancel out the divergent term in their paper.

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
