[Supplementary Material]

# A  Fokker-Planck Equation and Backward Kolmogorov Equation

In this section, we introduce the Fokker-Planck Equation and the Backward Kolmogorov equation. Fokker-Planck equation addresses the evolution of probability density $p(\mathbf{x})$ that associates with the SDE. We give the following specific definition.

**Definition A.1** (Fokker–Planck Equation). Let $p(\mathbf{x}, t)$ be the probability density at time $t$ of the stochastic differential equation and denote $p_0(\mathbf{x})$ the initial probability density. Then

$$\partial_t p(\mathbf{x}, t) = \mathcal{L}^* p(\mathbf{x}, t), \quad p(\mathbf{x}, 0) = p_0(\mathbf{x}),$$

where $\mathcal{L}^*$ is the formal adjoint of $\mathcal{L}$.

Fokker-Planck equation gives us a way to find whether there exists a stationary distribution for the SDE. It can be shown [30] that for the stochastic differential equation (1.2), its stationary distribution exists and satisfies

$$\pi(d\mathbf{x}) = \frac{1}{Q} e^{-\beta F_n(\mathbf{x})}, \quad Q = \int e^{-\beta F_n(\mathbf{x})} d\mathbf{x}. \tag{A.1}$$

This is also known as Gibbs measure.

Backward Kolmogorov equation describes the evolution of $\mathbb{E}[g(\mathbf{X}(t))|\mathbf{X}(0) = \mathbf{x}]$ with $g$ being a smooth test function.

**Definition A.2** (Backward Kolmogorov Equation). Let $\mathbf{X}(t)$ solves the stochastic differential equation (1.2). Let $u(\mathbf{x}, t) = \mathbb{E}[g(\mathbf{X}(t))|\mathbf{X}(0) = \mathbf{x}]$, we have

$$\partial_t u(\mathbf{x}, t) = \mathcal{L} u(\mathbf{x}, t), \quad u(\mathbf{x}, 0) = g(\mathbf{x}).$$

Now consider doing first order Taylor expansion on $u(\mathbf{x}, t)$, we have

$$
\begin{aligned}
u(\mathbf{x}, t) &= u(\mathbf{x}, 0) + \frac{\partial}{\partial t} u(\mathbf{x}, t)|_{t=0} \cdot (t - 0) + O(t^2) \\
&= g(\mathbf{x}) + t \mathcal{L} g(\mathbf{x}) + O(t^2).
\end{aligned} \tag{A.2}
$$

# B  Proof of Corollaries

In this section, we provide the proofs of corollaries for iteration complexity in our main theory section.

*Proof of Corollary 3.4.* To ensure the iterate error converge to $\epsilon$ precision, we need

$$\Theta e^{-\lambda K \eta} \leq \frac{\epsilon}{2}, \qquad \frac{C_\psi \eta}{\beta} \leq \frac{\epsilon}{2}.$$

The second inequality can be easily satisfied with $\eta = O(\epsilon)$ and the first inequality implies

$$K \geq \frac{1}{\lambda \eta} \log\left(\frac{2\Theta}{\epsilon}\right).$$

Combining with $\eta = O(\epsilon)$ and $\Theta = O(d^2/\rho^{d/2})$, we obtain the iteration complexity

$$K = O\left(\frac{d}{\epsilon \lambda} \cdot \log\left(\frac{1}{\epsilon}\right)\right),$$

which completes the proof.

$\square$

*Proof of Corollary 3.7.* To ensure the iterate error of SGLD converging to $\epsilon$ precision, we require the following inequalities to hold

$$C_1 \sqrt{\beta} \Gamma(M\sqrt{\Gamma} + G) K \eta \left[\frac{n - B}{B(n - 1)}\right]^{1/4} \leq \frac{\epsilon}{3}, \qquad \Theta e^{-\lambda K \eta} \leq \frac{\epsilon}{3}, \qquad \frac{C_\psi \eta}{\beta} \leq \frac{\epsilon}{3}.$$

The third inequality can be easily satisfied with $\eta = O(\epsilon)$. For the second inequality, similar as in the proof of Corollary 3.4, we have

$$K\eta \geq \frac{1}{\lambda} \log\left(\frac{3\Theta}{\epsilon}\right).$$

Since $\epsilon < 1$, we know that $\log(1/\epsilon)$ will not go to zero when $\epsilon$ goes to zero. In fact, if we set $\eta = O(\epsilon)$ and $K = O(d/(\lambda\epsilon) \log(1/\epsilon))$, the first term in (3.2) scales as

$$C_1\sqrt{\beta}\Gamma(M\sqrt{\Gamma} + G)K\eta\left[\frac{n-B}{B(n-1)}\right]^{1/4} = O\left(\frac{d^{3/2}K\eta}{B^{1/4}}\right) = O\left(\frac{d^{3/2}}{B^{1/4}\lambda} \log\left(\frac{1}{\epsilon}\right)\right).$$

Therefore, within $K = O(d/(\epsilon\lambda) \cdot \log(1/\epsilon))$ iterations, the iterate error of SGLD scales as

$$O\left(\frac{d^{3/2}}{B^{1/4}\lambda} \log\left(\frac{1}{\epsilon}\right) + \epsilon\right).$$

$\square$

*Proof of Corollary 3.11.* Similar to previous proofs, in order to achieve an $\epsilon$-precision iterate error for SVRG-LD, we require

$$C_1\Gamma K^{3/4}\eta\left[\frac{L\beta M^2(n-B)}{B(n-1)}\left(9\eta(M^2\Gamma + G^2) + \frac{d}{\beta}\right)\right]^{1/4} \leq \frac{\epsilon}{3}, \quad \Theta e^{-\lambda K\eta} \leq \frac{\epsilon}{3}, \quad \frac{C_\psi\eta}{\beta} \leq \frac{\epsilon}{3}.$$

By previous proofs we know that the second and third inequalities imply $\eta = O(\epsilon)$ and $K\eta = O(1/\lambda \log(3\Theta/\epsilon))$ respectively. Combining with the first inequality, we have

$$\eta^{1/4} = O\left(\frac{B^{1/4}\epsilon}{(K\eta)^{3/4}d^{5/4}L^{1/4}}\right)$$

Combining with the first inequality, we have

$$\eta = O\left(\min\left\{\frac{B\epsilon^4}{(K\eta)^3 d^5 L}, \epsilon\right\}\right)$$

Combining the above requirements yields

$$K = O\left(\frac{Ld^5}{B\lambda^4\epsilon^4} \log^4\left(\frac{1}{\epsilon}\right) + \frac{1}{\epsilon}\right). \tag{B.1}$$

For gradient complexity, note that for each iteration we need $B$ stochastic gradient evaluations and we also need in total $K/L$ full gradient calculations. Therefore, the gradient complexity for SVRG-LD is

$$O(K \cdot B + K/L \cdot n) = \widetilde{O}\left(\left(\frac{n}{B} + L\right)\frac{1}{\epsilon^4} + \left(\frac{n}{L} + B\right)\frac{1}{\epsilon}\right) \cdot e^{\widetilde{O}(d)}.$$

If we solve for the best $B$ and $L$, we obtain $B = \sqrt{n}\epsilon^{-3/2}$, $L = \sqrt{n}\epsilon^{3/2}$. Therefore, we have the optimal gradient complexity for SVRG-LD as

$$\widetilde{O}\left(\frac{\sqrt{n}}{\epsilon^{5/2}}\right) \cdot e^{\widetilde{O}(d)}.$$

$\square$

# C  Proof of Technical Lemmas

In this section, we provide proofs of the technical lemmas used in the proof of our main theory.

## C.1 Proof of Lemma 4.1

Geometric ergodicity of dynamical systems has been studied a lot in the literature [47, 40]. In particular, Roberts and Tweedie [47] proved that even when the diffusion converges exponentially fast to its stationary distribution, the Euler-Maruyama discretization in (2.2) may still lose the convergence properties and examples for Langevin diffusion can be found therein. To further address this problem, [40] built their analysis of ergodicity for SDEs on a *minorization* condition and the existence of a Lyapunov function. In time discretization of dynamics systems, they studied how time-discretization affects the minorization condition and the Lyapunov structure. For the self-containedness of our analysis, we present the minorization condition on a compact set $\mathcal{C}$ as follows.

**Proposition C.1.** There exist $t_0 \in \mathbb{R}$ and $\xi > 0$ such that the Markov process $\{\boldsymbol{X}(t)\}$ satisfies

$$\mathbb{P}(\boldsymbol{X}(t_0) \in A | \boldsymbol{X}(0) = \mathbf{x}) \geq \xi\nu(A),$$

for any $A \in \mathcal{B}(\mathbb{R}^d)$, some fixed compact set $\mathcal{C} \in \mathcal{B}(\mathbb{R}^d)$, and $\mathbf{x} \in \mathcal{C}$, where $\mathcal{B}(\mathbb{R}^d)$ denotes the Borel $\sigma$-algebra on $\mathbb{R}^d$ and $\nu$ is a probability measure with $\nu(\mathcal{C}^c) = 0$ and $\nu(\mathcal{C}) = 1$.

Proposition C.1 does not always hold for a Markov process generated by an arbitrary SDE. However, for Langevin diffusion (1.2) studied in this paper, Mattingly et al. [40] proved that this minorization condition actually holds under the dissipative and smooth assumptions (see Corollary 7.4 in Mattingly et al. [40]). For more explanation on the existence and robustness of the minorization condition under discretization approximations for Langevin diffusion, we refer interested readers to Corollary 7.5 and the proof of Theorem 6.2 in Mattingly et al. [40]. Now we are going to prove Lemma 4.1, which requires the following useful lemmas:

**Lemma C.2.** Let $V(\mathbf{x}) = C + \|\mathbf{x}\|_2^2$ be a function on $\mathbb{R}^d$, where $C > 0$ is a constant. Denote the expectation with Markov process $\{\boldsymbol{X}(t)\}$ starting at $\mathbf{x}$ by $\mathbb{E}^{\mathbf{x}}[\cdot] = \mathbb{E}[\cdot|\boldsymbol{X}(0) = \mathbf{x}]$. Under Assumption 3.2, we have

$$\mathbb{E}^{\mathbf{x}}[V(\boldsymbol{X}(t))] \leq e^{-2mt}V(\mathbf{x}) + \frac{b+m+d/\beta}{m}(1 - e^{-2mt}),$$

for all $\mathbf{x} \in \mathbb{R}^d$.

**Lemma C.3.** (Theorem 7.3 in Mattingly et al. [40]) Under Assumptions 3.1 and 3.2, let $V(\mathbf{x}) = C_0 + M/2\|\mathbf{x}\|_2^2$ be an essential quadratic function. The numerical approximation (2.1) (GLD) of Langevin diffusion (1.2) has a unique invariant measure $\mu$ and for all test function $g$ such that $|g| \leq V$, we have

$$\left|\mathbb{E}[g(\boldsymbol{X}_k)] - \mathbb{E}[g(\boldsymbol{X}^\mu)]\right| \leq C\kappa\rho^{-d/2}(1 + \kappa e^{m\eta}) \exp\left(-\frac{2mk\eta\rho^d}{\log(\kappa)}\right),$$

where $\rho \in (0,1), C > 0$ are absolute constants, and $\kappa = 2M(b+m+d)/m$.

*Proof of Lemma 4.1.* The proof is majorly adapted from that of Theorem 7.3 and Corollary 7.5 in Mattingly et al. [40]. By Assumption 3.1, $F_n$ is $M$-smooth. Thus we have

$$F_n(\mathbf{x}) \leq F_n(\mathbf{y}) + \langle \nabla F_n(\mathbf{y}), \mathbf{x} - \mathbf{y}\rangle + \frac{M}{2}\|\mathbf{x} - \mathbf{y}\|_2^2,$$

for all $\mathbf{x}, \mathbf{y} \in \mathbb{R}^d$. By Lemma D.1 and choosing $\mathbf{y} = \mathbf{0}$, this immediately implies that $F_n(\mathbf{x})$ can always be bounded by a quadratic function $V(\mathbf{x})$, i.e.,

$$F_n(\mathbf{x}) \leq \frac{M}{2}V(\mathbf{x}) = \frac{M}{2}(C_0 + \|\mathbf{x}\|_2^2).$$

Therefore $V(\mathbf{x})$ is an essentially quadratic Lyapunov function such that $|F_n(\mathbf{x})| \leq MV(\mathbf{x})/2$ for $\mathbf{x} \in \mathbb{R}^d$. By Lemma C.2 the Lyapunov function satisfies

$$\mathbb{E}^{\mathbf{x}_0}[V(\boldsymbol{X}(t))] \leq e^{-2mt}V(\mathbf{x}_0) + \frac{b+m+d/\beta}{m}(1 - e^{-2mt}).$$

According to Corollary 7.5 in Mattingly et al. [40], the Markov chain $\{\boldsymbol{X}_k\}_{k=1,2,\ldots,K}$ satisfies

$$\mathbb{E}^{\mathbf{x}_0}[MV(\boldsymbol{X}_1)/2] \leq e^{-2m\eta}[MV(\mathbf{x}_0)/2] + \frac{M(b+m+d/\beta)}{2m}. \tag{C.1}$$

Recall the GLD update formula defined in (2.1)

$$\boldsymbol{X}_{k+1} = \boldsymbol{X}_k - \eta \nabla F_n(\boldsymbol{X}_k) + \sqrt{2\eta\beta^{-1}} \cdot \boldsymbol{\epsilon}_k.$$

Define $F'(\boldsymbol{X}_k) = \beta F_n(\boldsymbol{X}_k)$ and $\eta' = \eta/\beta$, we have

$$\boldsymbol{X}_{k+1} = \boldsymbol{X}_k - \eta' \nabla F'(\boldsymbol{X}_k) + \sqrt{2\eta'} \cdot \boldsymbol{\epsilon}_k. \tag{C.2}$$

This suggests that the result for $\beta \neq 1$ is equivalent to rescaling $\eta$ to $\eta/\beta$ and $F_n(\cdot)$ to $\beta F_n(\cdot)$. Therefore, in the following proof, we will assume that $\beta = 1$ and then rescale $\eta$, $F_n(\cdot)$ at last. Similar tricks are used in Raginsky et al. [45], Zhang et al. [53]. Under Assumptions 3.1 and 3.2, it is proved that Euler-Maruyama approximation of Langevin dynamics (1.2) has a unique invariant measure $\mu$ on $\mathbb{R}^d$. Denote $\boldsymbol{X}^\mu$ as a random vector which is sampled from measure $\mu$. By Lemma C.3, for all test function $g$ such that $|g| \leq V$, it holds that

$$\left| \mathbb{E}[g(\boldsymbol{X}_k)] - \mathbb{E}[g(\boldsymbol{X}^\mu)] \right| \leq C\kappa'\rho^{-d/2}(1 + \kappa' e^{m\eta}) \exp\left( -\frac{2mk\eta\rho^d}{\log(\kappa')} \right),$$

where $\rho, \delta \in (0,1), C > 0$ are absolute constants, and $\kappa' = 2M(b + m + d)/m$. Take $F_n$ as the test function and $\boldsymbol{X}_0 = \boldsymbol{0}$, and by rescaling $\eta$ and $F_n(\cdot)$ (dissipative and smoothness parameters), we have

$$\left| \mathbb{E}[F_n(\boldsymbol{X}_k)] - \mathbb{E}[F_n(\boldsymbol{X}^\mu)] \right| \leq C\kappa\rho^{-d/2}(1 + \kappa e^{m\eta}) \exp\left( -\frac{2mk\eta\rho^d}{\log(\kappa)} \right),$$

where $\kappa = 2M(b\beta + m\beta + d)/m$. $\qquad\qquad\qquad\qquad\qquad\qquad\qquad\qquad\square$

## C.2 Proof of Lemma 4.2

To prove Lemma 4.2, we lay down the following supporting lemma, of which the derivation is inspired and adapted from Chen et al. [12].

**Lemma C.4.** Under Assumptions 3.1 and 3.2, the Markov chain $\{\boldsymbol{X}_k\}_{k=1}^K$ generated by Algorithm 1 satisfies

$$\left| \frac{1}{K} \sum_{k=0}^{K-1} \mathbb{E}[F_n(\boldsymbol{X}_k)|\boldsymbol{X}_0 = \mathbf{x}] - \bar{F} \right| \leq C_\psi \left( \frac{\beta}{\eta K} + \frac{\eta}{\beta} \right),$$

where $\bar{F} = \int F_n(\mathbf{x})\pi(d\mathbf{x})$ with $\pi$ being the Gibbs measure for the Langevin diffusion (1.2).

*Proof of Lemma 4.2.* By definition we have

$$\left| \mathbb{E}[F_n(\boldsymbol{X}^\mu)] - \mathbb{E}[F_n(\boldsymbol{X}^\pi)] \right| = \left| \int F_n(\mathbf{x})\mu(d\mathbf{x}) - \int F_n(\mathbf{x})\pi(d\mathbf{x}) \right|. \tag{C.3}$$

For simplicity, we denote the average $\int F_n(\mathbf{x})\pi(d\mathbf{x})$ as $\bar{F}_n$. Since $\mu$ is the ergodic limit of the Markov chain generated by the GLD process, for a given test function $F_n$, we have

$$\int F_n(\mathbf{x})\mu(d\mathbf{x}) = \int \mathbb{E}[F_n(\boldsymbol{X}_k)|\boldsymbol{X}_0 = \mathbf{x}] \cdot \mu(d\mathbf{x}).$$

Since $\mu$ and $\pi$ are two invariant measures, we consider the case where $K \to \infty$. Take average over $K$ steps $\{\boldsymbol{X}_k\}_{k=0}^{K-1}$ we have

$$\int F_n(\mathbf{x})\mu(d\mathbf{x}) = \lim_{K\to\infty} \int \frac{1}{K} \sum_{k=0}^{K-1} \mathbb{E}[F_n(\boldsymbol{X}_k)|\boldsymbol{X}_0 = \mathbf{x}] \cdot \mu(d\mathbf{x}). \tag{C.4}$$

Submitting (C.4) back into (C.3) yields

$$\left| \mathbb{E}[F_n(\boldsymbol{X}^\mu)] - \mathbb{E}[F_n(\boldsymbol{X}^\pi)] \right| = \lim_{K\to\infty} \left| \int \left[ \frac{1}{K} \sum_{k=0}^{K-1} \mathbb{E}[F_n(\boldsymbol{X}_k)|\boldsymbol{X}_0 = \mathbf{x}] - \bar{F} \right] \cdot \mu(d\mathbf{x}) \right|$$

$$\leq \lim_{K\to\infty} \int \left| \frac{1}{K} \sum_{k=0}^{K-1} \mathbb{E}[F_n(\boldsymbol{X}_k)|\boldsymbol{X}_0 = \mathbf{x}] - \bar{F} \right| \cdot \mu(d\mathbf{x}). \tag{C.5}$$

Apply Lemma C.4 with $g$ chosen as $F_n$ we further bound (C.5) by

$$\left| \mathbb{E}[F_n(\boldsymbol{X}^\mu)] - \mathbb{E}[F_n(\boldsymbol{X}^\pi)] \right| \leq C_\psi \cdot \lim_{K \to \infty} \int \left( \frac{\beta}{\eta K} + \frac{\eta}{\beta} \right) \cdot \mu(d\mathbf{x})$$

$$= C_\psi \cdot \lim_{K \to \infty} \left( \frac{\beta}{\eta K} + \frac{\eta}{\beta} \right)$$

$$= \frac{C_\psi \eta}{\beta}.$$

$\square$

## C.3 Proof of Lemma 4.4

Lemma 4.4 gives the upper bound of function value gap between the GLD iterates and the SGLD iterates. To bound the difference between $F_n(\boldsymbol{X}_K)$ and $F_n(\boldsymbol{Y}_K)$, we need the following lemmas.

**Lemma C.5.** Under Assumptions 3.1 and 3.2, for any $\mathbf{x} \in \mathbb{R}^d$, it holds that

$$\mathbb{E} \left\| \nabla F_n(\mathbf{x}) - \frac{1}{B} \sum_{i \in I_k} \nabla f_i(\mathbf{x}) \right\|_2^2 \leq \frac{4(n-B)(M\|\mathbf{x}\|_2 + G)^2}{B(n-1)},$$

where $B = |I_k|$ is the mini-batch size and $G = \max_{i=1,\dots,n}\{\|\nabla f_i(\mathbf{x}^*)\|_2\} + bM/m$.

The following lemma describes the $L^2$ bound for discrete processes $\boldsymbol{X}_k$ (GLD), $\boldsymbol{Y}_k$ (SGLD) and $\boldsymbol{Z}_k$ (SVRG-LD). Note that for SGLD, similar result is also presented in Raginsky et al. [45].

**Lemma C.6.** Under Assumptions 3.1 and 3.2, for sufficiently small step size $\eta$, suppose the initial points of Algorithms 1, 2 and 3 are chosen at $\mathbf{0}$, then the $L^2$ bound of the GLD process (2.1), SGLD process (2.2) and SVRG-LD process (2.3) can be uniformly bounded by

$$\max\{\mathbb{E}[\|\boldsymbol{X}_k\|_2^2], \mathbb{E}[\|\boldsymbol{Y}_k\|_2^2], \mathbb{E}[\|\boldsymbol{Z}_k\|_2^2]\} \leq \Gamma \quad \text{where} \quad \Gamma := 2\left(1 + \frac{1}{m}\right)\left(b + 2G^2 + \frac{d}{\beta}\right),$$

for any $k = 0, 1, \dots, K$, where $G = \max_{i=1,\dots,n}\{\|\nabla f_i(\mathbf{x}^*)\|_2\} + bM/m$.

The following lemma gives out the upper bound for the exponential $L^2$ bound of $\boldsymbol{X}_k$.

**Lemma C.7.** Under Assumptions 3.1 and 3.2, for sufficiently small step size $\eta < 1$ and the inverse temperature satisfying $\beta \geq \max\{2/(m - M^2\eta), 4\eta\}$, it holds that

$$\log \mathbb{E}[\exp(\|\boldsymbol{X}_k\|_2^2)] \leq \|\boldsymbol{X}_0\|_2^2 + \frac{2\beta(b+G^2) + 2d}{\beta - 4\eta} k\eta.$$

**Lemma C.8.** [44, 45] For any two probability density functions $\mu, \nu$ with bounded second moments, let $g : \mathbb{R}^d \to \mathbb{R}$ be a $C^1$ function such that

$$\|\nabla g(\mathbf{x})\|_2 \leq C_1 \|\mathbf{x}\|_2 + C_2, \forall \mathbf{x} \in \mathbb{R}^d$$

for some constants $C_1, C_2 \geq 0$. Then

$$\left| \int_{\mathbb{R}^d} g(\mathbf{x}) d\mu - \int_{\mathbb{R}^d} g(\mathbf{x}) d\nu \right| \leq (C_1 \sigma + C_2) \mathcal{W}_2(\mu, \nu),$$

where $\mathcal{W}_2$ is the 2-Wasserstein distance and $\sigma^2 = \max\left\{ \int_{\mathbb{R}^d} \|\mathbf{x}\|_2^2 \mu(d\mathbf{x}), \int_{\mathbb{R}^d} \|\mathbf{x}\|_2^2 \nu(d\mathbf{x}) \right\}$.

**Lemma C.9.** (Corollary 2.3 in Bolley and Villani [5]) Let $\nu$ be a probability measure on $\mathbb{R}^d$. Assume that there exist $\mathbf{x}_0$ and a constant $\alpha > 0$ such that $\int \exp(\alpha \|\mathbf{x} - \mathbf{x}_0\|_2^2) d\nu(\mathbf{x}) < \infty$. Then for any probability measure $\mu$ on $\mathbb{R}^d$, it satisfies

$$\mathcal{W}_2(\mu, \nu) \leq C_\nu \left( \sqrt{D_{\text{KL}}(\mu||\nu)} + \left( D_{\text{KL}}(\mu||\nu)/2 \right)^{1/4} \right),$$

where $C_\nu$ is defined as

$$C_\nu = \inf_{\mathbf{x}_0 \in \mathbb{R}^d, \alpha > 0} \sqrt{\frac{1}{\alpha}\left( \frac{3}{2} + \log \int \exp(\alpha \|\mathbf{x} - \mathbf{x}_0\|_2^2) d\nu(\mathbf{x}) \right)}.$$

*Proof of Lemma 4.4.* Let $P_K, Q_K$ denote the probability measures for GLD iterate $\boldsymbol{X}_K$ and SGLD iterate $\boldsymbol{Y}_K$ respectively. Applying Lemma C.8 to probability measures $P_K$ and $Q_K$ yields

$$\left| \mathbb{E}[F_n(\boldsymbol{Y}_K)] - \mathbb{E}[F_n(\boldsymbol{X}_K)] \right| \leq (C_1\sqrt{\Gamma} + C_2)\mathcal{W}_2(Q_K, P_K), \tag{C.6}$$

where $C_1, C_2 > 0$ are absolute constants and $\Gamma = 2(1 + 1/m)(b + 2G^2 + d/\beta)$ is the upper bound for both $\mathbb{E}[\|\boldsymbol{X}_k\|_2^2]$ and $\mathbb{E}[\|\boldsymbol{Y}_k\|_2^2]$ according to Lemma C.6. We further bound the $\mathcal{W}_2$ distance via the KL-divergence by Lemma C.9 as follows

$$\mathcal{W}_2(Q_K, P_K) \leq \Lambda(\sqrt{D_{\mathrm{KL}}(Q_K\|P_K)} + \sqrt[4]{D_{\mathrm{KL}}(Q_K\|P_K)}), \tag{C.7}$$

where $\Lambda = \sqrt{3/2 + \log \mathbb{E}_{P_K}[\exp(\|\boldsymbol{X}_K\|_2^2)]}$. Applying Lemma C.7 we obtain $\Lambda = \sqrt{(6 + 2\Gamma)K\eta}$. Therefore, we only need to bound the KL-divergence between density functions $P_K$ and $Q_K$. To this end, we introduce a continuous-time Markov process $\{\boldsymbol{D}(t)\}_{t\geq 0}$ to bridge the gap between diffusion $\{\boldsymbol{X}(t)\}_{t\geq 0}$ and its numerical approximation $\{\boldsymbol{X}_k\}_{k=0,1,\ldots,K}$. Define

$$d\boldsymbol{D}(t) = b(\boldsymbol{D}(t))dt + \sqrt{2\beta^{-1}}d\boldsymbol{B}(t), \tag{C.8}$$

where $b(\boldsymbol{D}(t)) = -\sum_{k=0}^{\infty} \nabla F(\boldsymbol{X}(\eta k))\mathbb{1}\{t \in [\eta k, \eta(k+1))\}$. Integrating (C.8) on interval $[\eta k, \eta(k+1))$ yields

$$\boldsymbol{D}(\eta(k+1)) = \boldsymbol{D}(\eta k) - \eta\nabla F(\boldsymbol{D}(\eta k)) + \sqrt{2\eta\beta^{-1}} \cdot \boldsymbol{\epsilon}_k,$$

where $\boldsymbol{\epsilon}_k \sim N(\boldsymbol{0}, \mathbf{I}_{d\times d})$. This implies that the distribution of random vector $(\boldsymbol{X}_1, \ldots, \boldsymbol{X}_K)$ is equivalent to that of $(\boldsymbol{D}(\eta), \ldots, \boldsymbol{D}(\eta K))$. Similarly, for $\boldsymbol{Y}_k$ we define

$$d\widetilde{\boldsymbol{M}}(t) = c(\widetilde{\boldsymbol{M}}(t))dt + \sqrt{2\beta^{-1}}d\boldsymbol{B}(t),$$

where the drift coefficient is defined as $c(\widetilde{\boldsymbol{M}}(t)) = -\sum_{k=0}^{\infty} g_k(\widetilde{\boldsymbol{M}}(\eta k))\mathbb{1}\{t \in [\eta k, \eta(k+1))\}$ and $g_k(\mathbf{x}) = 1/B \sum_{i\in I_k} \nabla f_i(\mathbf{x})$ is a mini-batch of the full gradient with $I_k$ being a random subset of $\{1, 2, \ldots, n\}$ of size $B$. Now we have that the distribution of random vector $(\boldsymbol{Y}_1, \ldots, \boldsymbol{Y}_K)$ is equivalent to that of $(\widetilde{\boldsymbol{M}}(\eta), \ldots, \widetilde{\boldsymbol{M}}(\eta K))$. However, the process $\widetilde{\boldsymbol{M}}(t)$ is not Markov due to the randomness of the stochastic gradient $g_k$. Therefore, we define the following Markov process which has the same one-time marginals as

$$d\boldsymbol{M}(t) = h(\boldsymbol{M}(t))dt + \sqrt{2\beta^{-1}}d\boldsymbol{B}(t), \tag{C.9}$$

where $h(\cdot) = -\mathbb{E}[g_k(\widetilde{\boldsymbol{M}}(\eta k))\,\mathbb{1}\{t \in [\eta k, \eta(k+1))\}|\widetilde{\boldsymbol{M}}(t) = \cdot]$ is the conditional expectation of the left end point of the interval which $\widetilde{\boldsymbol{M}}(t)$ lies in. Let $\mathbb{P}_t$ denote the distribution of $\boldsymbol{D}(t)$ and $\mathbb{Q}_t$ denote the distribution of $\boldsymbol{M}(t)$. By (C.8) and (C.9), the Radon-Nikodym derivative of $\mathbb{P}_t$ with respective to $\mathbb{Q}_t$ is given by the following Girsanov formula [38]

$$\frac{d\mathbb{P}_t}{d\mathbb{Q}_t}(\boldsymbol{M}) = \exp\left\{ \sqrt{\frac{\beta}{2}} \int_0^t (h(\boldsymbol{M}(s)) - b(\boldsymbol{M}(s)))^\top (d\boldsymbol{M}(s) - h(\boldsymbol{M}(s))ds) \right.$$
$$\left. - \frac{\beta}{4} \int_0^t \|h(\boldsymbol{M}(s)) - b(\boldsymbol{M}(s))\|_2^2 ds \right\}.$$

Since Markov processes $\{\boldsymbol{D}(t)\}_{t\geq 0}$ and $\{\boldsymbol{M}(t)\}_{t\geq 0}$ are constructed based on Markov chains $\boldsymbol{X}_k$ and $\boldsymbol{Y}_k$, by data-processing inequality the K-L divergence between $P_K$ and $Q_K$ can be bounded by

$$D_{KL}(Q_K\|P_K) \leq D_{KL}(\mathbb{Q}_{\eta K}\|\mathbb{P}_{\eta K})$$
$$= -\mathbb{E}\left[ \log\left( \frac{d\mathbb{P}_{\eta K}}{d\mathbb{Q}_{\eta K}}(\boldsymbol{M}) \right) \right]$$
$$= \frac{\beta}{4} \int_0^{\eta K} \mathbb{E}\big[\|h(\boldsymbol{M}(r)) - b(\boldsymbol{M}(r))\|_2^2\big] dr, \tag{C.10}$$

where in the last equality we used the fact that $d\boldsymbol{B}(t)$ follows Gaussian distribution independently for any $t \geq 0$. By definition, we know that both $h(\boldsymbol{M}(r))$ and $b(\boldsymbol{M}(r))$ are step functions when

$r \in [\eta k, \eta(k+1))$ for any $k$. This observation directly yields

$$\int_0^{\eta K} \mathbb{E}\big[\|h(\boldsymbol{M}(r)) - b(\boldsymbol{M}(r))\|_2^2\big] dr \leq \sum_{k=0}^{K-1} \int_{\eta k}^{\eta(k+1)} \mathbb{E}\big[\|g_k(\widetilde{\boldsymbol{M}}(\eta k)) - \nabla F_n(\widetilde{\boldsymbol{M}}(\eta k))\|_2^2\big] dr$$

$$= \eta \sum_{k=0}^{K-1} \mathbb{E}\big[\|g_k(\boldsymbol{Y}_k) - \nabla F_n(\boldsymbol{Y}_k)\|_2^2\big],$$

where the first inequality is due to Jensen's inequality and the convexity of function $\|\cdot\|^2$, and the last equality is due to the equivalence in distribution. By Lemmas C.5 and C.6, we further have

$$\int_0^{\eta K} \mathbb{E}\big[\|h(\boldsymbol{M}(r)) - b(\boldsymbol{M}(r))\|_2^2\big] dr \leq \frac{4\eta K(n-B)(M\Gamma + G)^2}{B(n-1)}. \tag{C.11}$$

Submitting (C.10) and (C.11) into (C.7), we have

$$\mathcal{W}_2(Q_K, P_K) \leq \Lambda\left(\sqrt{\frac{\beta\eta K(n-B)(M\Gamma + G)^2}{B(n-1)}} + \sqrt[4]{\frac{\beta\eta K(n-B)(M\Gamma + G)^2}{B(n-1)}}\right)$$

$$\leq \Lambda\sqrt{\frac{\beta\eta K\sqrt{n-B}(M\Gamma + G)^2}{\sqrt{B(n-1)}}}. \tag{C.12}$$

Combining (C.6) with (C.12), we obtain the expected function value gap between SGLD and GLD:

$$|\mathbb{E}[F(\boldsymbol{Y}_k)] - \mathbb{E}[F(\boldsymbol{X}_k)]| \leq C_1\Gamma\sqrt{K\eta}\left[\frac{\beta\eta K\sqrt{n-B}(M\sqrt{\Gamma} + G)^2}{\sqrt{B(n-1)}}\right]^{1/2},$$

where we adopt the fact that $K\eta > 1$ and assume that $C_1 \geq C_2$.

$\square$

## C.4   Proof of Lemma 4.5

Similar to the proof of Lemma 4.4, to bound the difference between $F_n(\boldsymbol{X}_K)$ and $F_n(\boldsymbol{Z}_K)$, we need the following lemmas.

**Lemma C.10.** Under Assumptions 3.1 and 3.2, for each iteration $k = sL + \ell$ in Algorithm 3, it holds that

$$\mathbb{E}\|\widetilde{\nabla}_k - \nabla F_n(\boldsymbol{Z}_k)\|_2^2 \leq \frac{M^2(n-B)}{B(n-1)}\mathbb{E}\|\boldsymbol{Z}_k - \widetilde{\boldsymbol{Z}}^{(s)}\|_2^2,$$

where $\widetilde{\nabla}_k = 1/B \sum_{i_k \in I_k} \left(\nabla f_{i_k}(\boldsymbol{Z}_k) - \nabla f_{i_k}(\widetilde{\boldsymbol{Z}}^{(s)}) + \nabla F_n(\widetilde{\boldsymbol{Z}}^{(s)})\right)$ and $B = |I_k|$ is the mini-batch size.

*Proof of Lemma 4.5.* Denote $Q_K^Z$ as the probability density functions for $\boldsymbol{Z}_K$. For the simplicity of notation, we omit the index $Z$ in the remaining part of this proof when no confusion arises. Similar as in the proof of Lemma 4.4, we first apply Lemma C.8 to probability measures $P_K$ for $\boldsymbol{X}_K$ and $Q_K^Z$ for $\boldsymbol{Z}_K$, and obtain the following upper bound of function value gap

$$|\mathbb{E}[F_n(\boldsymbol{Z}_K)] - \mathbb{E}[F_n(\boldsymbol{X}_K)]| \leq (C_1\sqrt{\Gamma} + C_2)\mathcal{W}_2(Q_K^Z, P_K), \tag{C.13}$$

where $C_1, C_2 > 0$ are absolute constants and $\Gamma = 2(1 + 1/m)(b + 2G^2 + d/\beta)$ is the upper bound for both $\mathbb{E}[\|\boldsymbol{X}_k\|_2^2]$ and $\mathbb{E}[\|\boldsymbol{Z}_k\|_2^2]$ according to Lemma C.6. Further by Lemma C.9, the $\mathcal{W}_2$ distance can be bounded by

$$\mathcal{W}_2(Q_K^Z, P_K) \leq \Lambda(\sqrt{D_{\mathrm{KL}}(Q_K^Z\|P_K)} + \sqrt[4]{D_{\mathrm{KL}}(Q_K^Z\|P_K)}), \tag{C.14}$$

where $\Lambda = \sqrt{3/2 + \log \mathbb{E}_{P_K}[e^{\|\boldsymbol{X}_K\|_2^2}]}$. Applying Lemma C.7 we obtain $\Lambda = \sqrt{(6 + 2\Gamma)K\eta}$. Therefore, we need to bound the KL-divergence between density functions $P_K$ and $Q_K^Z$. Similar to the proof of Lemma 4.4, we define a continuous-time Markov process associated with $\boldsymbol{Z}_k$ as follows

$$d\widetilde{\boldsymbol{N}}(t) = p(\widetilde{\boldsymbol{N}}(t))dt + \sqrt{2\beta^{-1}}d\boldsymbol{B}(t),$$

where $p(\widetilde{\boldsymbol{N}}(t)) = -\sum_{k=0}^{\infty} \widetilde{\nabla}_k \mathbb{1}\{t \in [\eta k, \eta(k+1))\}$ and $\widetilde{\nabla}_k$ is the semi-stochastic gradient at $k$-th iteration of SVRG-LD. We have that the distribution of random vector $(\boldsymbol{Z}_1, \ldots, \boldsymbol{Z}_K)$ is equivalent to that of $(\widetilde{\boldsymbol{N}}(\eta), \ldots, \widetilde{\boldsymbol{N}}(\eta K))$. However, $\widetilde{\boldsymbol{N}}(t)$ is not Markov due to the randomness of $\widetilde{\nabla}_k$. We define the following Markov process which has the same one-time marginals as $\widetilde{\boldsymbol{N}}(t)$

$$d\boldsymbol{N}(t) = q(\boldsymbol{N}(t))dt + \sqrt{2\beta^{-1}}d\boldsymbol{B}(t), \tag{C.15}$$

where $q(\cdot) = -\mathbb{E}[\widetilde{\nabla}_k \mathbb{1}\{t \in [\eta k, \eta(k+1))\}|p(\widetilde{\boldsymbol{N}}(t)) = \cdot]$. Let $\mathbb{Q}_t^Z$ denote the distribution of $\boldsymbol{N}(t)$. By (C.8) and (C.15), the Radon-Nikodym derivative of $\mathbb{P}_t$ with respective to $\mathbb{Q}_t^Z$ is given by the Girsanov formula [38]

$$\frac{d\mathbb{P}_t}{d\mathbb{Q}_t^Z}(\boldsymbol{N}) = \exp\left\{\sqrt{\frac{\beta}{2}}\int_0^t (q(\boldsymbol{N}(r)) - b(\boldsymbol{N}(r)))^\top (d\boldsymbol{N}(r) - h(\boldsymbol{N}(r))dr)\right.$$
$$\left. - \frac{\beta}{4}\int_0^t \|q(\boldsymbol{N}(r)) - b(\boldsymbol{N}(r))\|_2^2 dr\right\}.$$

Since Markov processes $\{\boldsymbol{D}(t)\}_{t\geq 0}$ and $\{\boldsymbol{N}(t)\}_{t\geq 0}$ are constructed based on $\boldsymbol{X}_k$ and $\boldsymbol{Z}_k$, by data-processing inequality the K-L divergence between $P_K$ and $Q_K^Z$ in (C.14) can be bounded by

$$D_{\text{KL}}(Q_K^Z \| P_K) \leq D_{\text{KL}}(\mathbb{Q}_{\eta K}^Z \| \mathbb{P}_{\eta K})$$
$$= -\mathbb{E}\left[\log\left(\frac{d\mathbb{P}_{\eta K}}{d\mathbb{Q}_{\eta K}^Z}(\boldsymbol{N})\right)\right]$$
$$= \frac{\beta}{4}\int_0^{\eta K} \mathbb{E}\left[\|q(\boldsymbol{N}(r)) - b(\boldsymbol{N}(r))\|_2^2\right]dr. \tag{C.16}$$

where in the last equality we used the fact that $d\boldsymbol{B}(t)$ follows Gaussian distribution independently for any $t \geq 0$. By definition, we know that both $q(\boldsymbol{N}(r))$ and $b(\boldsymbol{N}(r))$ are step functions when $r \in [\eta k, \eta(k+1))$ for any $k$. This observation directly yields

$$\int_0^{\eta K} \mathbb{E}\left[\|q(\boldsymbol{N}(r)) - b(\boldsymbol{N}(r))\|_2^2\right]dr \leq \sum_{k=0}^{K-1}\int_{\eta k}^{\eta(k+1)} \mathbb{E}\left[\widetilde{\nabla}_k(\widetilde{\boldsymbol{N}}(\eta k)) - \nabla F_n(\widetilde{\boldsymbol{N}}(\eta k))\|_2^2\right]dr$$
$$= \eta\sum_{k=0}^{K-1} \mathbb{E}\left[\|\widetilde{\nabla}_k(\boldsymbol{Z}_k) - \nabla F_n(\boldsymbol{Z}_k)\|_2^2\right],$$

where the first inequality is due to Jensen's inequality and the convexity of function $\|\cdot\|_2^2$, and the last equality is due to the equivalence in distribution. Combine the above results we obtain

$$D_{\text{KL}}(Q_K^Z \| P_K) \leq \frac{\beta\eta}{4}\sum_{k=0}^{K-1} \mathbb{E}[\|\widetilde{\nabla}_k - \nabla F_n(\boldsymbol{Z}_k)\|_2^2]$$
$$\leq \frac{\beta\eta}{4}\sum_{s=0}^{K/L}\sum_{\ell=0}^{L-1} \mathbb{E}[\|\widetilde{\nabla}_{sL+\ell} - \nabla F_n(\boldsymbol{Z}_{sL+\ell})\|_2^2], \tag{C.17}$$

where the second inequality follows the fact that $k = sL+\ell \leq (s+1)L$ for some $\ell = 0, 1, \ldots, L-1$. Applying Lemma C.10, the inner summation in (C.17) yields

$$\sum_{\ell=0}^{L-1} \mathbb{E}[\|\widetilde{\nabla}_{sL+\ell} - \nabla F_n(\boldsymbol{Z}_{sL+\ell})\|_2^2] \leq \sum_{\ell=0}^{L-1} \frac{M^2(n-B)}{B(n-1)}\mathbb{E}\|\boldsymbol{Z}_{sL+\ell} - \widetilde{\boldsymbol{Z}}^{(s)}\|_2^2. \tag{C.18}$$

Note that we have

$$\mathbb{E}\|\boldsymbol{Z}_{sL+\ell} - \widetilde{\boldsymbol{Z}}^{(s)}\|_2^2$$
$$= \mathbb{E}\left\|\sum_{u=0}^{\ell-1} \eta(\nabla f_{i_{sL+u}}(\boldsymbol{Z}_{sL+u}) - \nabla f_{i_{sL+u}}(\widetilde{\boldsymbol{Z}}^{(s)}) + \nabla F_n(\widetilde{\boldsymbol{Z}}^{(s)})) - \sum_{u=0}^{\ell-1}\sqrt{\frac{2\eta}{\beta}}\epsilon_{sL+\ell}\right\|_2^2$$
$$\leq \ell\sum_{u=0}^{\ell-1} \mathbb{E}\left[2\eta^2\|\nabla f_{i_{sL+u}}(\boldsymbol{Z}_{sL+u}) - \nabla f_{i_{sL+u}}(\widetilde{\boldsymbol{Z}}^{(s)}) + \nabla F_n(\widetilde{\boldsymbol{Z}}^{(s)})\|_2^2\right] + \sum_{u=0}^{\ell-1}\frac{4\eta d}{\beta}$$
$$\leq 4\ell\eta\left(9\ell\eta(M^2\Gamma^2 + G^2) + \frac{d}{\beta}\right), \tag{C.19}$$

where the first inequality holds due to the triangle inequality for the first summation term, the second one follows from Lemma D.1 and Lemma C.6. Submit (C.19) back into (C.18) we have

$$\sum_{\ell=0}^{L-1} \mathbb{E}[\|\widetilde{\nabla}_{sL+\ell} - \nabla F_n(\boldsymbol{Z}_{sL+\ell})\|_2^2] \leq \frac{4\eta M^2(n-B)}{B(n-1)} \sum_{\ell=0}^{L-1} \left( 9\ell^2 \eta (M^2\Gamma^2 + G^2) + \frac{\ell d}{\beta} \right)$$

$$\leq \frac{4\eta M^2(n-B)}{B(n-1)} \left( 3L^3 \eta (M^2\Gamma + G^2) + \frac{dL^2}{2\beta} \right), \quad \text{(C.20)}$$

Since (C.20) does not depend on the outer loop index $i$, submitting it into (C.17) yields

$$\frac{\beta\eta}{4} \sum_{k=0}^{K-1} \mathbb{E}[\|\widetilde{\nabla}_k - \nabla F_n(\boldsymbol{Z}_k)\|_2^2] \leq \frac{\eta^2 KLM^2(n-B)(3L\eta\beta(M^2\Gamma + G^2) + d/2)}{B(n-1)}. \quad \text{(C.21)}$$

Combining (C.13), (C.14) (C.17) and (C.21), we obtain

$$\left| \mathbb{E}[F_n(\boldsymbol{Z}_K)] - \mathbb{E}[F_n(\boldsymbol{X}_K)] \right| \leq C_1 \Gamma \sqrt{K\eta} \left[ \frac{\eta^2 KLM^2(n-B)(3L\eta\beta(M^2\Gamma + G^2) + d/2)}{B(n-1)} \right]^{1/4}.$$

where we use the fact that $K\eta > 1$, $\eta < 1$ and assume that $C_1 \geq C_2$. $\qquad \square$

# D  Proof of Auxiliary Lemmas

In this section, we prove additional lemmas used in Appendix C.

## D.1  Proof of Lemma C.2

*Proof.* Applying Itô's Lemma yields

$$dV(\boldsymbol{X}(t)) = -2\langle \boldsymbol{X}(t), \nabla F_n(\boldsymbol{X}(t))\rangle dt + \frac{2d}{\beta} dt + 2\sqrt{\frac{2}{\beta}} \langle \boldsymbol{X}(t), d\boldsymbol{B}(t)\rangle. \quad \text{(D.1)}$$

Multiplying $e^{2mt}$ to both sides of the above equation, where $m > 0$ is the dissipative constant, we obtain

$$2me^{2mt}V(\boldsymbol{X}(t))dt + e^{2mt}dV(\boldsymbol{X}(t)) = 2me^{2mt}V(\boldsymbol{X}(t))dt - 2e^{2mt}\langle \boldsymbol{X}(t), \nabla F_n(\boldsymbol{X}(t))\rangle dt$$

$$+ \frac{2d}{\beta} e^{2mt}dt + \sqrt{\frac{8}{\beta}} e^{2mt}\langle \boldsymbol{X}(t), d\boldsymbol{B}(t)\rangle.$$

We integrate the above equation from time $0$ to $t$ and have

$$V(\boldsymbol{X}(t)) = e^{-2mt}V(\boldsymbol{X}_0) + 2m \int_0^t e^{2m(s-t)}V(\boldsymbol{X}(s))ds - 2\int_0^t e^{2m(s-t)}\langle \boldsymbol{X}(s), \nabla F_n(\boldsymbol{X}(s))\rangle ds$$

$$+ \frac{2d}{\beta} \int_0^t e^{2m(s-t)}ds + 2\sqrt{\frac{2}{\beta}} \int_0^t e^{2m(s-t)}\langle \boldsymbol{X}(s), d\boldsymbol{B}(s)\rangle. \quad \text{(D.2)}$$

Note that by Assumption 3.2, we have

$$-2\int_0^t e^{2m(s-t)}\langle \boldsymbol{X}(s), \nabla F_n(\boldsymbol{X}(s))\rangle ds \leq -2\int_0^t e^{2m(s-t)}\left( m\|\boldsymbol{X}(s)\|_2^2 - b\right)ds$$

$$= -2m \int_0^t e^{2m(s-t)}V(\boldsymbol{X}(s))ds + \frac{b+m}{m}(1 - e^{-2mt}). \quad \text{(D.3)}$$

Combining (D.2) and (D.3), and taking expectation over $\boldsymbol{X}(t)$ with initial point $\mathbf{x}$, we get

$$\mathbb{E}^{\mathbf{x}}[V(\boldsymbol{X}(t))] \leq e^{-2mt}V(\mathbf{x}) + \frac{b+m}{m}(1 - e^{-2mt}) + \frac{d}{m\beta}(1 - e^{-2mt})$$

$$= e^{-2mt}V(\mathbf{x}) + \frac{b+m+d/\beta}{m}(1 - e^{-2mt}),$$

where we employed the fact that $d\boldsymbol{B}(s)$ follows Gaussian distribution with zero mean and is independent with $\boldsymbol{X}(s)$. $\qquad \square$

## D.2    Proof of Lemma C.3

Here we provide a sketch of proof to refine the parameters in the results of Mattingly et al. [40]. For detailed proof, we refer interested readers to Theorem 7.3 in Mattingly et al. [40].

*Proof.* Denote $\kappa = 2M(b + m + d)/m$ according to Lemma C.2 where $b, m$ are the dissipative parameters. We also define $\phi = \rho^d$ with some constant $0 < \rho < 1$. Let $\{\boldsymbol{X}_{l\tau}\}_{l=0,1,...}$ be a sub-sampled chain from $\{\boldsymbol{X}_k\}_{k=0,1,...}$ at sample rate $\tau > 0$. By the proof of Theorem 2.5 in Mattingly et al. [40], we obtain the following result

$$\left|\mathbb{E}[g(\boldsymbol{X}_{l\tau})] - \mathbb{E}[g(\boldsymbol{X}^\mu)]\right| \leq \kappa[\bar{V} + 1](1 - \phi)^{\alpha l\tau} + \sqrt{2}V(\mathbf{x}_0)\delta^{l\tau}\kappa^{\alpha l\tau/2}\frac{1}{\sqrt{\phi}}, \quad \text{(D.4)}$$

where $\boldsymbol{X}^\mu$ follows the invariant distribution of Markov process $\{\boldsymbol{X}_k\}_{k=0,1,...}$, $\bar{V} = 2\sup_{\mathbf{x}\in\mathcal{C}} V(\mathbf{x})$ is a bounded constant, $\delta \in (e^{-2m\eta}, 1)$ is a constant, and $\alpha \in (0, 1)$ is chosen small enough such that $\delta\kappa^{\alpha/2} \leq 1$. In particular, we choose $\alpha \in (0, 1)$ such that $\delta\kappa^{\alpha/2} \leq (1 - \phi)^\alpha$, which yields

$$\alpha \leq \frac{\log(1/\delta)}{\log(\sqrt{\kappa}/(1 - \phi))} \leq \frac{\log(1/\delta)}{\log(\sqrt{\kappa})},$$

where the last inequality is due to $1 - \phi < 1$. Submitting the choice of $\alpha$ into (D.4) we have

$$\left|\mathbb{E}[g(\boldsymbol{X}_{l\tau})] - \mathbb{E}[g(\boldsymbol{X}^\mu)]\right| \leq \frac{2\sqrt{2}\kappa}{\sqrt{\phi}}[\bar{V} + 1]V(\mathbf{x}_0)(1 - \phi)^{l\tau \log(1/\delta)/\log(\sqrt{\kappa})}$$

$$= \frac{2\sqrt{2}\kappa}{\sqrt{\phi}}[\bar{V} + 1]V(\mathbf{x}_0)e^{l\tau \log(r)}, \quad \text{(D.5)}$$

where $r = (1 - \phi)^{\log(1/\delta)/\log(\sqrt{\kappa})}$ is defined as the contraction parameter. Note that by Taylor's expansion we have

$$\log r = \log(1 - (1 - r)) = -(1 - r) - \frac{(1 - r)^2}{2} - \frac{(1 - r)^3}{3} - \ldots \leq -(1 - r), \quad \text{(D.6)}$$

when $|1 - r| \leq 1$. By definition $r = (1 - \phi)^{\log(1/\delta)/\log(\sqrt{\kappa})}$ and $\phi = \rho^d$ where $\rho \in (0, 1)$ is a constant. Since it is more interesting to deal with the situation where dimension parameter $d$ is large enough and not negligible, we can always assume that $|\phi| = \rho^d$ is sufficiently small such that for any $0 < \zeta < 1$

$$(1 - \phi)^\zeta = 1 - \zeta\phi + \zeta(\zeta - 1)/2\phi^2 + \ldots + \binom{\zeta}{n}(-\phi)^n + \ldots \leq 1 - \zeta\phi \quad \text{(D.7)}$$

by Taylor's expansion. Submitting (D.6) and (D.7) into (D.5) yields

$$\left|\mathbb{E}[g(\boldsymbol{X}_{l\tau})] - \mathbb{E}[g(\boldsymbol{X}^\mu)]\right| \leq \frac{2\sqrt{2}\kappa}{\sqrt{\phi}}[\bar{V} + 1]V(\mathbf{x}_0)\exp\left(-\frac{2ml\tau\eta\rho^d}{\log(\kappa)}\right), \quad \text{(D.8)}$$

where we chose $\delta = e^{-m\eta}$. Next we need to prove that the unsampled chain is also exponential ergodic. Let $k = l\tau + j$ with $j = 0, 1, \ldots, \tau - 1$. We immediately get

$$\left|\mathbb{E}[g(\boldsymbol{X}_{l\tau+j})] - \mathbb{E}[g(\boldsymbol{X}^\mu)]\right| \leq \frac{2\sqrt{2}\kappa}{\sqrt{\phi}}[\bar{V} + 1]\mathbb{E}[V(\boldsymbol{X}_j)]\exp\left(-\frac{2ml\tau\eta\rho^d}{\log(\kappa)}\right).$$

Since the GLD approximation (2.1) of Langevin is ergodic when sampled at rate $\tau = 1$, we have $k = l\tau = l$ and $j = 0$. Note that by Lemma A.2 in Mattingly et al. [40], we have $\mathcal{C} = \{\mathbf{x} : V(\mathbf{x}) \leq \kappa/e^{-m\eta}\}$, which implies that $\bar{V} = \kappa e^{m\eta}$. Thus we obtain

$$\left|\mathbb{E}[g(\boldsymbol{X}_k)] - \mathbb{E}[g(\boldsymbol{X}^\mu)]\right| \leq C\kappa\rho^{-d/2}(\kappa e^{m\eta} + 1)\exp\left(-\frac{2mk\eta\rho^d}{\log(\kappa)}\right),$$

where we used the fact that $\mathbf{x}_0 = \mathbf{0}$ and $C > 0$ is an absolute constant. $\square$

## D.3 Proof of Lemma C.4

To prove Lemma C.4, we choose the test function in Poisson equation (1.3) as $g = F_n$. Given the Poisson equation, suppose we choose $g$ as $F_n$, the distance between the time average of the GLD process and the expectation of $F_n$ over the Gibbs measure can be expressed by

$$\frac{1}{K}\sum_{k=1}^{K}F_n(\boldsymbol{X}_k) - \bar{F} = \frac{1}{K}\sum_{k=1}^{K}\mathcal{L}\psi(\boldsymbol{X}_k). \tag{D.9}$$

Note that by [41, 51], we know the Poisson equation (1.3) defined by the generator of Langevin dynamics has a unique solution $\psi$ under Assumptions 3.1 and 3.2. According to Theorem 3.2 in [23], the $p$-th order derivatives of $\psi$ can be bounded by some polynomial growth function with sophisticated coefficients ($p = 0, 1, 2$). To simplify the presentation, we hence follow the convention in the line of literature [12, 51] and assume that $\mathbb{E}\left[\|\nabla^p\psi(\boldsymbol{X}_k)\|\right]$ can be further upper bounded by a constant $C_\psi$ for all $\{\boldsymbol{X}_k\}_{k\geq 0}$ and $p = (0, 1, 2)$, which is determined by the Langevin diffusion and its Poisson equation. In fact, Erdogdu et al. [23] showed that the upper bound of derivatives (up to fourth order) of $\psi$ only requires the dissipative and smooth assumptions. We refer interested readers to [23] for more details on deriving the $C_\psi$ for Langevin diffusion. We show that the case $p = 0$ can be easily verified as follows. By Assumption 3.1, using a similar argument as in the proof of Lemma 4.1, we bound $F_n(\mathbf{x})$ by a quadratic function $V(\mathbf{x})$

$$F_n(\mathbf{x}) \leq \frac{M}{2}V(\mathbf{x}) = \frac{M}{2}(C_0 + \|\mathbf{x}\|_2^2).$$

Applying Assumption 3.2 and Theorem 13 in Vollmer et al. [51] we have

$$|\psi(\mathbf{x})| \leq C_1(1 + \|\mathbf{x}\|_2^2) \leq C_2 V(\mathbf{x}). \tag{D.10}$$

Note that by Assumptions 3.1 and 3.2 we can verify that a quadratic $V(\mathbf{x})$ and $p^* = 2$ satisfy Assumption 12 in [51] and therefore we obtain that for all $p \leq p^*$, we have

$$\sup_k \mathbb{E}V^p(\boldsymbol{X}_k) \leq \infty. \tag{D.11}$$

Combining (D.10) and (D.11) we show that $\psi(\boldsymbol{X}_k)$ is bounded in expectation.

*Proof.* For the simplicity of notation, we first assume that $\beta = 1$ and then show the result for arbitrary $\beta$ by a scaling technique. Note that for the continuous-time Markov process $\{\boldsymbol{D}(t)\}_{t\geq 0}$ defined in (C.8), the distribution of random vector $(\boldsymbol{X}_1, \ldots, \boldsymbol{X}_K)$ is equivalent to that of $(\boldsymbol{D}(\eta), \ldots, \boldsymbol{D}(\eta K))$. Let $\psi$ be the solution of Poisson equation $\mathcal{L}\psi = g - \int g(\mathbf{x})\pi(d\mathbf{x})$. Since we have $\mathbb{E}[\psi(\boldsymbol{X}_k)|\boldsymbol{X}_0 = \mathbf{x}] = \mathbb{E}[\psi(\boldsymbol{D}(\eta k))|\boldsymbol{D}_0 = \mathbf{x}]$. We denote $\mathbb{E}[\psi(\boldsymbol{D}(\eta k))|\boldsymbol{D}_0 = \mathbf{x}]$ by $\mathbb{E}^{\mathbf{x}}[\psi(\boldsymbol{D}(\eta k))]$. By applying (A.2), we compute the Taylor expansion of $\mathbb{E}^{\mathbf{x}}[\psi(\boldsymbol{D}(\eta k))]$ at $\boldsymbol{D}(\eta(k-1))$:

$$\mathbb{E}^{\mathbf{x}}[\psi(\boldsymbol{D}(\eta k))] = \mathbb{E}^{\mathbf{x}}[\psi(\boldsymbol{D}(\eta(k-1)))] + \eta\mathbb{E}^{\mathbf{x}}[\mathcal{L}\psi(\boldsymbol{D}(\eta(k-1)))] + O(\eta^2).$$

Note that the remainder also depends on the second order derivative of the Poisson equation and are bounded by constant $C_\psi$. Take average over $k = 1, \ldots, K$ and rearrange the equation we have

$$\frac{1}{\eta K}\left(\mathbb{E}^{\mathbf{x}}[\psi(\boldsymbol{D}(\eta K))] - \psi(\mathbf{x})\right) + O(\eta) = \frac{1}{K}\sum_{k=1}^{K}\mathbb{E}^{\mathbf{x}}[\mathcal{L}\psi(\boldsymbol{D}(\eta(k-1)))]. \tag{D.12}$$

Submit the Poisson equation (D.9) into the above equation (D.12) we have

$$\begin{aligned}
\frac{1}{K}\sum_{k=0}^{K-1}\mathbb{E}^{\mathbf{x}}[F_n(\boldsymbol{X}_k)] - \bar{F} &= \frac{1}{K}\sum_{k=1}^{K}\mathbb{E}^{\mathbf{x}}[\mathcal{L}\psi(\boldsymbol{X}_{k-1})] = \frac{1}{K}\sum_{k=1}^{K}\mathbb{E}^{\mathbf{x}}[\mathcal{L}\psi(\boldsymbol{D}(\eta(k-1)))] \\
&= \frac{1}{\eta K}\left(\mathbb{E}^{\mathbf{x}}[\psi(\boldsymbol{D}(\eta K))] - \psi(\mathbf{x})\right) + O(\eta) \\
&= \frac{1}{\eta K}\left(\mathbb{E}^{\mathbf{x}}[\psi(\boldsymbol{X}_K)] - \psi(\mathbf{x})\right) + O(\eta),
\end{aligned}$$

where the second and the fourth equation hold due to the fact that the distribution of $\{\boldsymbol{X}_k\}$ is the same as the distribution of $\{\boldsymbol{D}(\eta k)\}$. We have assumed that $\psi(\boldsymbol{X}_k)$ and its first and second order

derivatives are bounded by constant $C_\psi$ in expectation over the randomness of $\boldsymbol{X}_k$. Therefore, we are able to obtain the following conclusion

$$\left| \frac{1}{K} \sum_{k=0}^{K-1} \mathbb{E}^{\mathbf{x}}[F_n(\boldsymbol{X}_k)] - \bar{F} \right| \leq C_\psi \left( \frac{1}{\eta K} + \eta \right).$$

This completes the proof for the case $\beta = 1$. In order to apply our analysis to the case where $\beta$ can take any arbitrary constant value, we conduct the same scaling argument as in (C.2).

$$\left| \frac{1}{K} \sum_{k=0}^{K-1} \mathbb{E}^{\mathbf{x}}[F_n(\boldsymbol{X}_k)] - \bar{F} \right| \leq C_\psi \left( \frac{1}{\eta' K} + \eta' \right) = C_\psi \left( \frac{\beta}{\eta K} + \frac{\eta}{\beta} \right).$$

This completes the proof. $\qquad\square$

### D.4 Proof of Lemma C.5

We first lay down the following lemma on the bounds of gradient of $f_i$.

**Lemma D.1.** For any $\mathbf{x} \in \mathbb{R}^d$, it holds that

$$\|\nabla f_i(\mathbf{x})\|_2 \leq M\|\mathbf{x}\|_2 + G$$

for constant $G = \max_{i=1,\ldots,n}\{\|\nabla f_i(\mathbf{x}^*)\|_2\} + bM/m$.

*Proof of Lemma C.5.* Let $\mathbf{u}_i(\mathbf{x}) = \nabla F(\mathbf{x}) - \nabla f_i(\mathbf{x})$, consider

$$\begin{aligned}
\mathbb{E}\left\| \frac{1}{B} \sum_{i \in I_k} \mathbf{u}_i(\mathbf{x}) \right\|_2^2 &= \frac{1}{B^2} \mathbb{E} \sum_{i \neq i' \in I_k} \mathbf{u}_i(\mathbf{x})^\top \mathbf{u}_{i'}(\mathbf{x}) + \frac{1}{B} \mathbb{E}\|\mathbf{u}_i(\mathbf{x})\|_2^2 \\
&= \frac{B-1}{Bn(n-1)} \sum_{i \neq i'} \mathbf{u}_i(\mathbf{x})^\top \mathbf{u}_{i'}(\mathbf{x}) + \frac{1}{B} \mathbb{E}\|\mathbf{u}_i(\mathbf{x})\|_2^2 \\
&= \frac{B-1}{Bn(n-1)} \sum_{i,i'} \mathbf{u}_i(\mathbf{x})^\top \mathbf{u}_{i'}(\mathbf{x}) - \frac{B-1}{B(n-1)} \mathbb{E}\|\mathbf{u}_i(\mathbf{x})\|_2^2 + \frac{1}{B} \mathbb{E}\|\mathbf{u}_i(\mathbf{x})\|_2^2 \\
&= \frac{n-B}{B(n-1)} \mathbb{E}\|\mathbf{u}_i(\mathbf{x})\|_2^2, \qquad\qquad\qquad\qquad\qquad\qquad\text{(D.13)}
\end{aligned}$$

where the last equality is due to the fact that $1/n \sum_{i=1}^n \mathbf{u}_i(\mathbf{x}) = 0$. By Lemma D.1 we have $\|\nabla f_i(\mathbf{x})\|_2 \leq M\|\mathbf{x}\|_2 + G$, therefore we have $\|\nabla F(\mathbf{x})\|_2 \leq M\|\mathbf{x}\|_2 + G$ and consequently, $\|\mathbf{u}_i(\mathbf{x})\|_2 \leq 2(M\|\mathbf{x}\|_2 + G)$. Thus (D.13) can be further bounded as:

$$\mathbb{E}\left\| \frac{1}{B} \sum_{i \in I_k} \mathbf{u}_i(\mathbf{x}) \right\|_2^2 \leq \frac{n-B}{B(n-1)} 4(M\|\mathbf{x}\|_2 + G)^2.$$

This completes the proof. $\qquad\square$

### D.5 Proof of Lemma C.6

In this section, we provide the proof of $L^2$ bound of GLD and SVRG-LD iterates $\boldsymbol{X}_k$ and $\boldsymbol{Z}_k$. Note that a similar result of SGLD has been proved in Raginsky et al. [45] and thus we omit the corresponding proof for the simplicity of presentation.

*Proof of Lemma C.6.* **Part I**: We first prove the the upper bound for GLD. By the definition in (2.1), we have

$$\begin{aligned}
\mathbb{E}[\|\boldsymbol{X}_{k+1}\|_2^2] &= \mathbb{E}[\|\boldsymbol{X}_k - \eta \nabla F_n(\boldsymbol{X}_k)\|_2^2] + \sqrt{\frac{8\eta}{\beta}} \mathbb{E}[\langle \boldsymbol{X}_k - \eta \nabla F_n(\boldsymbol{X}_k), \boldsymbol{\epsilon}_k \rangle] + \frac{2\eta}{\beta} \mathbb{E}[\|\boldsymbol{\epsilon}_k\|_2^2] \\
&= \mathbb{E}[\|\boldsymbol{X}_k - \eta \nabla F_n(\boldsymbol{X}_k)\|_2^2] + \frac{2\eta d}{\beta},
\end{aligned}$$

where the second equality follows from that $\epsilon_k$ is independent on $\boldsymbol{X}_k$. Now we bound the first term

$$\mathbb{E}[\|\boldsymbol{X}_k - \eta\nabla F_n(\boldsymbol{X}_k)\|_2^2] = \mathbb{E}[\|\boldsymbol{X}_k\|_2^2] - 2\eta\mathbb{E}[\langle\boldsymbol{X}_k, \nabla F_n(\boldsymbol{X}_k)\rangle] + \eta^2\mathbb{E}[\|\nabla F_n(\boldsymbol{X}_k)\|_2^2]$$
$$\leq \mathbb{E}[\|\boldsymbol{X}_k\|_2^2] + 2\eta(b - m\mathbb{E}[\|\boldsymbol{X}_k\|_2^2]) + 2\eta^2(M^2\mathbb{E}[\|\boldsymbol{X}_k\|_2^2] + G^2)$$
$$= (1 - 2\eta m + 2\eta^2 M^2)\mathbb{E}[\|\boldsymbol{X}_k\|_2^2] + 2\eta b + 2\eta^2 G^2,$$

where the inequality follows from Assumption 3.2, Lemma D.1 and triangle inequality. Substitute the above bound back and we will have

$$\mathbb{E}[\|\boldsymbol{X}_{k+1}\|_2^2] \leq (1 - 2\eta m + 2\eta^2 M^2)\mathbb{E}[\|\boldsymbol{X}_k\|_2^2] + 2\eta b + 2\eta^2 G^2 + \frac{2\eta d}{\beta}. \qquad (\text{D.14})$$

For sufficient small $\eta$ that satisfies $\eta \leq \min\{1, m/(2M^2)\}$, there are only two cases we need to take into account:
If $1 - 2\eta m + 2\eta^2 M^2 \leq 0$, then from (D.14) we have

$$\mathbb{E}[\|\boldsymbol{X}_{k+1}\|_2^2] \leq 2\eta b + 2\eta^2 G^2 + \frac{2\eta d}{\beta} \leq \|\boldsymbol{X}_0\|_2^2 + 2\left(b + G^2 + \frac{d}{\beta}\right). \qquad (\text{D.15})$$

If $0 < 1 - 2\eta m + 2\eta^2 M^2 \leq 1$, then iterate (D.14) and we have

$$\mathbb{E}[\|\boldsymbol{X}_k\|_2^2] \leq (1 - 2\eta m + 2\eta^2 M^2)^k\|\boldsymbol{X}_0\|_2^2 + \frac{\eta b + \eta^2 G^2 + \frac{\eta d}{\beta}}{\eta m - \eta^2 M^2} \leq \|\boldsymbol{X}_0\|_2^2 + \frac{2}{m}\left(b + G^2 + \frac{d}{\beta}\right). \tag{D.16}$$

Combine (D.15) and (D.16) and we have

$$\mathbb{E}[\|\boldsymbol{X}_k\|_2^2] \leq \|\boldsymbol{X}_0\|_2^2 + \left(2 + \frac{2}{m}\right)\left(b + G^2 + \frac{d}{\beta}\right) = 2\left(1 + \frac{1}{m}\right)\left(b + G^2 + \frac{d}{\beta}\right),$$

where the equation holds by choosing $\boldsymbol{X}_0 = \boldsymbol{0}$.

**Part II**: Now we prove the $L^2$ bound for SVRG-LD, i.e., $\mathbb{E}[\|\boldsymbol{Z}_k\|_2^2]$, by mathematical induction. Since $\widetilde{\nabla}_k = 1/B \sum_{i_k \in I_k} \left(\nabla f_{i_k}(\boldsymbol{Z}_k) - \nabla f_{i_k}(\widetilde{\boldsymbol{Z}}^{(s)}) + \nabla F_n(\widetilde{\boldsymbol{Z}}^{(s)})\right)$, we have

$$\mathbb{E}[\|\boldsymbol{Z}_{k+1}\|_2^2] = \mathbb{E}[\|\boldsymbol{Z}_k - \eta\widetilde{\nabla}_k\|_2^2] + \sqrt{\frac{8\eta}{\beta}}\mathbb{E}[\langle\boldsymbol{Z}_k - \eta\widetilde{\nabla}_k, \epsilon_k\rangle] + \frac{2\eta}{\beta}\mathbb{E}[\|\epsilon_k\|_2^2]$$
$$= \mathbb{E}[\|\boldsymbol{Z}_k - \eta\widetilde{\nabla}_k\|_2^2] + \frac{2\eta d}{\beta}, \qquad (\text{D.17})$$

where the second equality follows from the fact that $\epsilon_k$ is independent of $\boldsymbol{Z}_k$ and standard Gaussian. We prove it by induction. First, consider the case when $k = 1$. Since we choose the initial point at $\boldsymbol{Z}_0 = \boldsymbol{0}$, we immediately have

$$\mathbb{E}[\|\boldsymbol{Z}_1\|_2^2] = \mathbb{E}[\|\boldsymbol{Z}_0 - \eta\widetilde{\nabla}_0\|_2^2] + \sqrt{\frac{8\eta}{\beta}}\mathbb{E}[\langle\boldsymbol{Z}_0 - \eta\widetilde{\nabla}_0, \epsilon_0\rangle] + \frac{2\eta}{\beta}\mathbb{E}[\|\epsilon_0\|_2^2]$$
$$= \eta^2\mathbb{E}[\|\nabla F_n(\boldsymbol{Z}_0)\|_2^2] + \frac{2\eta d}{\beta}$$
$$\leq \eta^2 G^2 + \frac{2\eta d}{\beta},$$

where the second equality holds due to the fact that $\widetilde{\nabla}_0 = \nabla F_n(\boldsymbol{Z}_0)$ and the inequality follows from Lemma D.1. For sufficiently small $\eta$ we can see that the conclusion of Lemma C.6 holds for $\mathbb{E}[\|\boldsymbol{Z}_1\|_2^2]$, i.e., $\mathbb{E}[\|\boldsymbol{Z}_1\|_2^2] \leq \Gamma$, where $\Gamma = 2(1 + 1/m)(b + 2G^2 + d/\beta)$. Now assume that the conclusion holds for all iteration from 1 to $k$, then for the $(k+1)$-th iteration, by (D.17) we have,

$$\mathbb{E}[\|\boldsymbol{Z}_{k+1}\|_2^2] = \mathbb{E}[\|\boldsymbol{Z}_k - \eta\widetilde{\nabla}_k\|_2^2] + \frac{2\eta d}{\beta}, \qquad (\text{D.18})$$

For the first term on the R.H.S of (D.18) we have

$$\mathbb{E}[\|\boldsymbol{Z}_k - \eta\widetilde{\nabla}_k\|_2^2] = \mathbb{E}[\|\boldsymbol{Z}_k - \eta\nabla F_n(\boldsymbol{Z}_k)\|_2^2] + 2\eta\mathbb{E}\langle\boldsymbol{Z}_k - \eta\nabla F_n(\boldsymbol{Z}_k), \nabla F_n(\boldsymbol{Z}_k) - \widetilde{\nabla}_k\rangle$$
$$+ \eta^2\mathbb{E}[\|\nabla F_n(\boldsymbol{Z}_k) - \widetilde{\nabla}_k\|_2^2]$$
$$= \underbrace{\mathbb{E}[\|\boldsymbol{Z}_k - \eta\nabla F_n(\boldsymbol{Z}_k)\|_2^2]}_{T_1} + \underbrace{\eta^2\mathbb{E}[\|\nabla F_n(\boldsymbol{Z}_k) - \widetilde{\nabla}_k\|_2^2]}_{T_2}, \qquad (\text{D.19})$$

where the second equality holds due to the fact that $\mathbb{E}[\widetilde{\nabla}_k] = \nabla F_n(\boldsymbol{Z}_k)$. For term $T_1$, we can further bound it by

$$\begin{aligned}
\mathbb{E}[\|\boldsymbol{Z}_k - \eta \nabla F_n(\boldsymbol{Z}_k)\|_2^2] &= \mathbb{E}[\|\boldsymbol{Z}_k\|_2^2] - 2\eta \mathbb{E}[\langle \boldsymbol{Z}_k, \nabla F_n(\boldsymbol{Z}_k)\rangle] + \eta^2 \mathbb{E}[\|\nabla F_n(\boldsymbol{Z}_k)\|_2^2] \\
&\leq \mathbb{E}[\|\boldsymbol{Z}_k\|_2^2] + 2\eta(b - m\mathbb{E}[\|\boldsymbol{Z}_k\|_2^2]) + 2\eta^2(M^2\mathbb{E}[\|\boldsymbol{Z}_k\|_2^2] + G^2) \\
&= (1 - 2\eta m + 2\eta^2 M^2)\mathbb{E}[\|\boldsymbol{Z}_k\|_2^2] + 2\eta b + 2\eta^2 G^2,
\end{aligned}$$

where the inequality follows from Lemma D.1 and triangle inequality. For term $T_2$, by Lemma C.10 we have

$$\mathbb{E}\|\nabla F_n(\boldsymbol{Z}_k) - \widetilde{\nabla}_k\|_2^2 \leq \frac{M^2(n-B)}{B(n-1)}\mathbb{E}\|\boldsymbol{Z}_k - \widetilde{\boldsymbol{Z}}^{(s)}\|_2^2 \leq \frac{2M^2(n-B)}{B(n-1)}\Big(\mathbb{E}\|\boldsymbol{Z}_k\|_2^2 + \mathbb{E}\|\widetilde{\boldsymbol{Z}}^{(s)}\|_2^2\Big).$$

Submit the above bound back into (D.17) we have

$$\begin{aligned}
\mathbb{E}[\|\boldsymbol{Z}_{k+1}\|_2^2] \leq{}& \left(1 - 2\eta m + 2\eta^2 M^2\Big(1 + \frac{n-B}{B(n-1)}\Big)\right)\mathbb{E}[\|\boldsymbol{Z}_k\|_2^2] \\
&+ \frac{2\eta^2 M^2(n-B)}{B(n-1)}\mathbb{E}\|\widetilde{\boldsymbol{Z}}^{(s)}\|_2^2 + 2\eta b + 2\eta^2 G^2 + \frac{2\eta d}{\beta}. \quad\text{(D.20)}
\end{aligned}$$

Note that by assumption we have $\mathbb{E}\|\boldsymbol{Z}_j\|_2^2 \leq \Gamma$ for all $j = 1, \ldots, k$ where $\Gamma = 2(1 + 1/m)(b + 2G^2 + d/\beta)$, thus (D.20) can be further bounded as:

$$\mathbb{E}[\|\boldsymbol{Z}_{k+1}\|_2^2] \leq \underbrace{\left(1 - 2\eta m + 2\eta^2 M^2\Big(1 + \frac{2(n-B)}{B(n-1)}\Big)\right)}_{C_\lambda}\Gamma + 2\eta b + 2\eta^2 G^2 + \frac{2\eta d}{\beta}. \quad\text{(D.21)}$$

For sufficient small $\eta$ that satisfies

$$\eta \leq \min\left(1, \frac{m}{2M^2(1 + 2(n-B)/(B(n-1)))}\right),$$

there are only two cases we need to take into account:
If $C_\lambda \leq 0$, then from (D.21) we have

$$\mathbb{E}[\|\boldsymbol{Z}_{k+1}\|_2^2] \leq 2\eta b + 2\eta^2 G^2 + \frac{2\eta d}{\beta} \leq 2\left(b + G^2 + \frac{d}{\beta}\right). \quad\text{(D.22)}$$

If $0 < C_\lambda \leq 1$, then iterate (D.21) and we have

$$\mathbb{E}[\|\boldsymbol{Z}_{k+1}\|_2^2] \leq C_\lambda^{k+1}\|\boldsymbol{Z}_0\|_2^2 + \frac{\eta b + \eta^2 G^2 + \frac{\eta d}{\beta}}{\eta m - \eta^2 M^2\Big(1 + \frac{2(n-B)}{B(n-1)}\Big)} \leq \frac{2}{m}\left(b + G^2 + \frac{d}{\beta}\right). \quad\text{(D.23)}$$

Combining (D.22) and (D.23), we have

$$\mathbb{E}[\|\boldsymbol{Z}_{k+1}\|_2^2] \leq 2\left(1 + \frac{1}{m}\right)\left(b + 2G^2 + \frac{d}{\beta}\right).$$

Thus we show that when $\mathbb{E}[\|\boldsymbol{Z}_j\|_2^2], j = 1, \ldots, k$ are bounded, $\mathbb{E}[\|\boldsymbol{Z}_{k+1}\|_2^2]$ is also bounded. By mathematical induction we complete the proof. □

### D.6  Proof of Lemma C.7

*Proof.* We have the following equation according to the update of GLD in (2.1),

$$\begin{aligned}
\mathbb{E}[\exp(\|\boldsymbol{X}_{k+1}\|_2^2)] &= \mathbb{E}\exp\left(\left\|\boldsymbol{X}_k - \eta \nabla F_n(\boldsymbol{X}_k) + \sqrt{\frac{2\eta}{\beta}}\boldsymbol{\epsilon}_k\right\|_2^2\right) \\
&= \mathbb{E}\exp\left(\|\boldsymbol{X}_k - \eta \nabla F_n(\boldsymbol{X}_k)\|_2^2 + \sqrt{\frac{8\eta}{\beta}}\langle \boldsymbol{X}_k - \eta \nabla F_n(\boldsymbol{X}_k), \boldsymbol{\epsilon}_k\rangle + \frac{2\eta}{\beta}\|\boldsymbol{\epsilon}_k\|_2^2\right).
\end{aligned}$$
$$\text{(D.24)}$$

Let $H(\mathbf{x}) = \exp(\|\mathbf{x}\|_2^2)$, we have $\mathbb{E}[H(\boldsymbol{X}_{k+1})] = \mathbb{E}_{\boldsymbol{X}_k}[\mathbb{E}[H(\boldsymbol{X}_{k+1})|\boldsymbol{X}_k]]$. Thus we can first compute the conditional expectation on the R.H.S of (D.24) given $\boldsymbol{X}_k$, then compute the expectation with respect to $\boldsymbol{X}_k$. Note that $\boldsymbol{\epsilon}_k$ follows standard multivariate normal distribution, i.e., $\boldsymbol{\epsilon}_k \sim N(\mathbf{0}, \mathbf{I}_{d \times d})$. Then it can be shown that

$$\mathbb{E}\left[\exp\left(\sqrt{\frac{8\eta}{\beta}}\langle \boldsymbol{X}_k - \eta \nabla F_n(\boldsymbol{X}_k), \boldsymbol{\epsilon}_k\rangle + \frac{2\eta}{\beta}\|\boldsymbol{\epsilon}_k\|_2^2\right)\bigg| \boldsymbol{X}_k\right]$$
$$= \frac{1}{(1 - 4\eta/\beta)^{d/2}}\exp\left(\frac{4\eta}{\beta - 4\eta}\|\boldsymbol{X}_k - \eta \nabla F_n(\boldsymbol{X}_k)\|_2^2\right)$$

holds as long as $\beta > 4\eta$. Plugging the above equation into (D.24), we have

$$\mathbb{E}[H(\boldsymbol{X}_{k+1})] = \frac{1}{(1 - 4\eta/\beta)^{d/2}}\mathbb{E}_{\boldsymbol{X}_k}\left[\exp\left(\frac{\beta}{\beta - 4\eta}\|\boldsymbol{X}_k - \eta \nabla F_n(\boldsymbol{X}_k)\|_2^2\right)\right]. \qquad (D.25)$$

Note that by Assumption 3.2 and Lemma D.1 we have

$$\mathbb{E}_{\boldsymbol{X}_k}\exp\left(\frac{\beta}{\beta - 4\eta}\|\boldsymbol{X}_k - \eta \nabla F_n(\boldsymbol{X})\|_2^2\right)$$
$$= \mathbb{E}_{\boldsymbol{X}_k}\exp\left(\frac{\beta}{\beta - 4\eta}\left(\|\boldsymbol{X}_k\|_2^2 - 2\eta\langle\boldsymbol{X}_k, \nabla F_n(\boldsymbol{X}_k)\rangle + \eta^2\|\nabla F_n(\boldsymbol{X}_k)\|_2^2\right)\right)$$
$$\leq \mathbb{E}_{\boldsymbol{X}_k}\exp\left(\frac{\beta}{\beta - 4\eta}\left(\|\boldsymbol{X}_k\|_2^2 - 2\eta(m\|\boldsymbol{X}_k\|_2^2 - b) + 2\eta^2(M^2\|\boldsymbol{X}_k\|_2^2 + G^2)\right)\right)$$
$$= \mathbb{E}_{\boldsymbol{X}_k}\exp\left(\frac{\beta}{\beta - 4\eta}\left((1 - 2\eta m + 2\eta^2 M^2)\|\boldsymbol{X}_k\|_2^2 + 2b\eta + 2\eta^2 G^2\right)\right).$$

Consider sufficiently small $\eta$ such that $\eta < m/M^2$. Then for $\beta$ satisfying $\beta \geq \max\{2/(m - M^2\eta), 4\eta\}$, we have $\beta(1 - 2\eta m + 2\eta^2 M^2)/(\beta - 4\eta) \leq 1$. Therefore, the above expectation can be upper bounded by

$$\mathbb{E}_{\boldsymbol{X}_k}\exp\left(\frac{\beta}{\beta - 4\eta}\|\boldsymbol{X}_k - \eta \nabla F_n(\boldsymbol{X})\|_2^2\right) \leq \exp\left(\frac{2\eta\beta(b + \eta G^2)}{\beta - 4\eta}\right)\mathbb{E}[H(\boldsymbol{X}_k)].$$

Substituting the above inequality into (D.25), it follows that

$$\mathbb{E}[H(\boldsymbol{X}_{k+1})] \leq \frac{1}{(1 - 4\eta/\beta)^{d/2}}\exp\left(\frac{2\eta\beta(b + \eta G^2)}{\beta - 4\eta}\right)\mathbb{E}[H(\boldsymbol{X}_k)]$$
$$\leq \exp\left(\frac{2\eta(\beta b + \eta\beta G^2 + d)}{\beta - 4\eta}\right)\mathbb{E}[H(\boldsymbol{X}_k)],$$

where we used the fact that $\log(1/(1 - x)) \leq x/(1 - x)$ for $0 < x < 1$ and that

$$\log\left(\frac{1}{(1 - 4\eta/\beta)^{d/2}}\right) = \frac{d}{2}\log\left(\frac{1}{1 - 4\eta/\beta}\right) \leq \frac{2d\eta/\beta}{1 - 4\eta/\beta} = \frac{2\eta d}{\beta - 4\eta}.$$

Then we are able to show by induction that

$$\mathbb{E}[H(\boldsymbol{X}_k)] \leq \exp\left(\frac{2k\eta(\beta b + \eta\beta G^2 + d)}{\beta - 4\eta}\right)\mathbb{E}[H(\|\boldsymbol{X}_0\|_2)],$$

which immediately implies that

$$\log\mathbb{E}[\exp(\|\boldsymbol{X}_k\|_2^2)] \leq \|\boldsymbol{X}_0\|_2^2 + \frac{2\beta(b + G^2) + 2d}{\beta - 4\eta}k\eta,$$

where we assume that $\eta \leq 1$ and $\beta > 4\eta$.

$\square$

### D.7 Proof of Lemma C.10

*Proof.* Since by Algorithm 3 we have $\widetilde{\nabla}_k = (1/B)\sum_{i_k\in I_k}\left(\nabla f_{i_k}(\boldsymbol{Z}_k)-\nabla f_{i_k}(\widetilde{\boldsymbol{Z}}^{(s)})+\nabla F_n(\widetilde{\boldsymbol{Z}}^{(s)})\right)$, therefore,

$$\mathbb{E}[\|\widetilde{\nabla}_k - \nabla F_n(\boldsymbol{Z}_k)\|_2^2] = \mathbb{E}\left\|\frac{1}{B}\sum_{i_k\in I_k}\left(\nabla f_{i_k}(\boldsymbol{Z}_k) - \nabla f_{i_k}(\widetilde{\boldsymbol{Z}}^{(s)}) + \nabla F_n(\widetilde{\boldsymbol{Z}}^{(s)}) - \nabla F_n(\boldsymbol{Z}_k)\right)\right\|_2^2.$$

Let $\mathbf{u}_i = \nabla F_n(\boldsymbol{Z}_k) - \nabla F_n(\widetilde{\boldsymbol{Z}}^{(s)}) - \left(\nabla f_{i_k}(\boldsymbol{Z}_k) - \nabla f_{i_k}(\widetilde{\boldsymbol{Z}}^{(s)})\right)$.

$$
\begin{aligned}
\mathbb{E}\left\|\frac{1}{B}\sum_{i\in I_k}\mathbf{u}_i(\mathbf{x})\right\|_2^2 &= \frac{1}{B^2}\mathbb{E}\sum_{i\neq i'\in I_k}\mathbf{u}_i(\mathbf{x})^\top\mathbf{u}_{i'}(\mathbf{x}) + \frac{1}{B}\mathbb{E}\|\mathbf{u}_i(\mathbf{x})\|_2^2 \\
&= \frac{B-1}{Bn(n-1)}\sum_{i\neq i'}\mathbf{u}_i(\mathbf{x})^\top\mathbf{u}_{i'}(\mathbf{x}) + \frac{1}{B}\mathbb{E}\|\mathbf{u}_i(\mathbf{x})\|_2^2 \\
&= \frac{B-1}{Bn(n-1)}\sum_{i,i'}\mathbf{u}_i(\mathbf{x})^\top\mathbf{u}_{i'}(\mathbf{x}) - \frac{B-1}{B(n-1)}\mathbb{E}\|\mathbf{u}_i(\mathbf{x})\|_2^2 + \frac{1}{B}\mathbb{E}\|\mathbf{u}_i(\mathbf{x})\|_2^2 \\
&= \frac{n-B}{B(n-1)}\mathbb{E}\|\mathbf{u}_i(\mathbf{x})\|_2^2, \qquad\qquad (\text{D.26})
\end{aligned}
$$

where the last equality is due to the fact that $1/n\sum_{i=1}^n\mathbf{u}_i(\mathbf{x}) = 0$. Therefore, we have

$$
\begin{aligned}
\mathbb{E}[\|\widetilde{\nabla}_k - \nabla F_n(\boldsymbol{Z}_k)\|_2^2] &\leq \frac{n-B}{B(n-1)}\mathbb{E}\|\mathbf{u}_i\|_2^2 \\
&= \frac{n-B}{B(n-1)}\mathbb{E}\|\nabla f_{i_k}(\boldsymbol{Z}_k) - \nabla f_{i_k}(\widetilde{\boldsymbol{Z}}) - \mathbb{E}[\nabla f_{i_k}(\boldsymbol{Z}_k) - \nabla f_{i_k}(\widetilde{\boldsymbol{Z}})]\|_2^2 \\
&\leq \frac{n-B}{B(n-1)}\mathbb{E}\|\nabla f_{i_k}(\boldsymbol{Z}_k) - \nabla f_{i_k}(\widetilde{\boldsymbol{Z}})\|_2^2 \\
&\leq \frac{M^2(n-B)}{B(n-1)}\mathbb{E}\|\boldsymbol{Z}_k - \widetilde{\boldsymbol{Z}}\|_2^2, \qquad\qquad (\text{D.27})
\end{aligned}
$$

where the second inequality holds due to the fact that $\mathbb{E}[\|\mathbf{x} - \mathbb{E}[\mathbf{x}]\|_2^2] \leq \mathbb{E}[\|\mathbf{x}\|_2^2]$ and the last inequality follows from Assumption 3.1. This completes the proof. $\qquad\square$

## E   Proof of Auxiliary Lemmas in Appendix D

### E.1   Proof of Lemma D.1

*Proof.* By Assumption 3.2 we obtain

$$\langle\mathbf{x}^*, \nabla F_n(\mathbf{x}^*)\rangle \geq m\|\mathbf{x}^*\|_2^2 - b.$$

Note that $\mathbf{x}^*$ is the minimizer for $F_n$, which implies that $\nabla F_n(\mathbf{x}^*) = \mathbf{0}$ and threfore $\|\mathbf{x}^*\|_2 \leq b/m$. By Assumption 3.1 we further have

$$\|\nabla f_i(\mathbf{x})\|_2 \leq \|\nabla f_i(\mathbf{x}^*)\|_2 + M\|\mathbf{x} - \mathbf{x}^*\|_2 \leq \|\nabla f_i(\mathbf{x}^*)\|_2 + \frac{bM}{m} + M\|\mathbf{x}\|_2.$$

The proof is completed by setting $G = \max_{i=1,\dots,n}\{\|\nabla f_i(\mathbf{x}^*)\|_2\} + bM/m$. $\qquad\square$