[Reviews · NeurIPS 2018]

Reviewer 1



The authors analyze ULA, SGLD and SVRGLD for nonconvex optimisation. They show that all the algorithms converge globally to an 'almost minimizer.' They use alternative proof techniques to the paper by Raginsky et al. (2018), which looks at a similar problem. This enables them to get better complexity bounds; SVRGLD is also not considered by Raginsky et al. (2018), which the authors show can give better complexity results in certain situations. I found the paper very easy to read and well presented, which I appreciate is hard to do with a theoretical paper such as this. While the paper follows quite closely to Raginsky et al. (2018), the author makes the novel contributions clear. The new proof method, improved complexity results, and analysis of SVRGLD seem to me to have considerable impact; though I admit I am more familiar with the sampling literature than the optimisation literature. I have a few minor points I might suggest: - I think more discussion of the dissipative assumption would be useful for the reader. Some intuition for which models will satisfy this assumption or not would be useful. - I thought the 'proof roadmap' was a nice way of presenting the theoretical ideas in a digestible way. Maybe clarifying what is meant by a 'sufficiently small stepsize' in Lemma 4.1? Is there an exact bound for this? - The footnote ^2 in the abstract looks a little like a squared term, perhaps that footnote can be left to the comma or end of the sentence? - I'm not sure its needed to state all 3 algorithms (ULA, SGLD and SVRGLD) in their own algorithm boxes, since the descriptions already contain many of the details and it takes up a lot of space. Perhaps SVRGLD is the only necessary algorithm statement since it is less well known?

Reviewer 2



After the author rebuttal: Thank you for your detailed and satisfying responses to all of my comments. I keep my overall score and congratulate you for a very nice paper that I enjoyed reading. The authors propose a new framework to analyze the global convergence of Langevin dynamics based algorithms in a nonconvex optimization context. Three type of algorithms are considered, including gradient Langevin Dynamics, its stochastic version and, finally, a stochastic version that incorporates variance reduction. The key idea is a novel decomposition of the optimization error that directly analyses the ergodicity of the numerical approximation of the Langevin diffusion, rather than measuring the error of the true dynamics and its discretization, which is a slowly converging term. Moreover, focusing on the numerical approximation bypasses the analysis of the convergence of the true dynamics to its stationary distribution. Bypassing these two steps, the authors are able to show improved convergence rates compared to previous work for gradient Langevin dynamics and its stochastic version. In addition, the authors provide a first global convergence guarantee for the variance reduced version in this nonconvex setting. They show that, under certain conditions, the variance reduced version is superior to the other two. My comments: 1. line 51: I understand this as you mean that you take the discrete-time diffusion to be the "ground truth" and do subsequent analysis based on this? If so there should be no discretization error, i.e. you bypass it by construction? What do you mean with "get around to a large extent"? 2. You refer to sigma2 as the "variance of the stochastic gradient". The variance of the gradient should formally be a matrix (variance-covariance) matrix since the gradient is a vector. Or is it another variance metric you have in mind? 3. In general, how does sigma2 (however you define it) scale with d and n? 4. Does the rate d^7 (or d^5 after variance reduction) mean that the stochastic versions are basically non-feasible for high-dimensional datasets? I am not an expert in optimization methods, but the consensus seem to be that optimization methods scale much better with respect to the dimension of the problem compared to say simulation methods. These rates with respect to d seem to suggest otherwise. 5. Do you have any empirical results to validate your mathematical derivations? Note that this is the reason that I responded "Not confident" on the reproducibility question. 6. You have assumed that eta is a constant "step-size". You could probably improve your rates a lot by clever step-size choices, for example, using second order methods. I suppose that you would have to make more assumptions and your results would be less general, but it would be interesting to hear your thoughts on how to extend this work beyond a constant step-size. 7. Should there be dependence on n in the rates for gradient Langevin dynamics in Section 1.1? Other comments: This paper is very well written, in particular considering that it is purely theoretical. The authors have done an excellent job in presenting the work in a comprehensible way (using 8 pages only!).

Reviewer 3



After the rebuttal: Thank you for your answers. In this paper, the authors analyze the global convergence behavior of Langevin-dynamics-based algorithms, namely GLD (aka ULA), SGLD, and variance-reduced-SGLD. The paper and the proofs are very well written and clear. The authors provide error bounds that are tighter than the ones that were recently established in Raginsky et al 2017. The proof strategy consists of first bounding the error of GLD, then bounding the error of the other two algorithms by relating them to GLD. For the error bounds of GLD, the authors follow a different path compared to Raginsky et al, and this way they are able to prove tighter bounds. Overall, I think the improved bounds do not bring any practicality if we compare them with Raginsky et al, since the improvements are still dominated by the exponential dependence on the dimension. In this sense, the paper has very limited practical importance. Moreover, I believe that some remarks, e.g. Remark 3.9 are not realistic. On the other hand, the theoretical analysis can be seen rather straightforward. For GLD, the error decomposition is quite a standard one and the individual bounds for the terms (that are obtained after decomposition) can be obtained by slightly extending existing results, e.g. using a scaling argument with Mattingly et al and Chen et al, then using Raginsky et al directly. For the other methods, the results are also obtained by using very standard techniques, mainly Girsanov's theorem. However, I still think that the paper is a good contribution to the community since it contains a unified analysis framework and slightly better bounds. I believe the framework would be useful for other practitioners/theoreticians. I only have minor comments: * The second footnote in the abstract is confusing since it looks like a square. * Line 254: typo in nevertheless * The authors should explicitly mention that Lemma 4.3 is coming from Raginsky et al, like they do in Lemma C3. * The authors assume that the Poisson equation has a solution. Is it implied by the assumptions? If not they should explicitly mention it. In any case this points needs more clarification. * There are new papers on variance reduction in SG-MCMC. The authors should cite them as well: Stochastic Variance-Reduced Hamilton Monte Carlo Methods Stochastic Gradient Hamiltonian Monte Carlo with Variance Reduction for Bayesian Inference On the Theory of Variance Reduction for Stochastic Gradient Monte Carlo * The authors might also be interested in citing this recent paper: Asynchronous Stochastic Quasi-Newton MCMC for Non-Convex Optimization